



# Deciphering the metamorphic evolution of the Pulo do Lobo metasedimentary belt (SW Iberian Variscides)

Irene Pérez-Cáceres[1], David Jesús Martínez Poyatos[1], Olivier Vidal[2], Olivier Beyssac[3], Fernando Nieto[4], José Fernando Simancas[1], Antonio Azor[1] and Franck Bourdelle[5]

Departamento de Geodinámica, Facultad de Ciencias, Universidad de Granada, Campus de Fuentenueva s/n, 18071 Granada, Spain.
Institut de Sciences de la Terre (ISTerre), CNRS-University of Grenoble 1, 1381 rue de la Piscine, 38041 Grenoble, France.
Institut de Physique des Matériaux et de Cosmochimie (IMPMC), CNRS-Sorbonne Université, Case Courrier 115, 4 place Jussieu, 75005 Paris, France.
Departamento de Mineralogía y Petrología, IACT, Facultad de Ciencias, Universidad de Granada-CSIC, Campus de Fuentenueva s/n, 18071 Granada, Spain.
Laboratoire Génie Civil et géo-Environnement (LGCgE), Université de Lille, Bât. SN5, Cité Scientifique, 59655 Villeneuve d'Ascq, France

Correspondence to: Irene Pérez-Cáceres (perezcaceres@ugr.es)

**Abstract**

The Pulo do Lobo belt is one of the units related to the orogenic suture between the Ossa-Morena and the South Portuguese zones in the SW Iberian Variscides. This metasedimentary unit has been classically interpreted as a Rheic subduction-related accretionary prism formed during the pre-Carboniferous convergence and eventual collision between the South Portuguese Zone (part of Avalonia) and the Ossa-Morena Zone (peri-Gondwanan terrane). Discrete mafic intrusions also occur in the dominant Pulo do Lobo metapelites, related to an intraorogenic Mississippian transtensional and magmatic event that had a significant thermal input. Three different approaches have been applied to the Devonian/Carboniferous phyllites and slates of the Pulo do Lobo belt in order to study their poorly known low-grade metamorphic evolution. X-Ray diffraction (XRD) was used to unravel the mineralogy and measure crystallographic parameters (illite "crystallinity" and K-white mica *b*-cell dimension). Compositional maps of selected samples were obtained from electron probe microanalysis, which allowed processing with XmapTools software, and chlorite semi-empirical and thermodynamic geothermometry was performed. Thermometry





based on Raman spectroscopy of carbonaceous material (RSCM) was used to obtain peak temperatures.

The microstructural study shows the existence of two phyllosilicate growth events at the chlorite zone, the main one ($M_1$) related to the development of a Devonian foliation $S_1$, and a minor one ($M_2$) associated with a crenulation cleavage ($S_2$) developed at middle/upper Carboniferous time. $M_1$ entered well into epizone (greenschist facies) conditions. $M_2$ conditions were at lower temperature, reaching the anchizone/epizone boundary. These data accord well with the unconformity that separates the Devonian and Carboniferous formations of the Pulo do Lobo belt. The varied results obtained by the different approaches followed, combined with microstructural analysis, are indicative of different snapshots of the metamorphic history. Thus, RSCM temperatures are higher in comparison with the other methods applied, which is interpreted as reflecting a faster reequilibration during the short-lived thermal Mississippian event. Regarding the metamorphic pressure, the data are very homogeneous (very low celadonite content in muscovite and low values of K-white mica $b$-cell dimension), indicating a low-pressure gradient, which is unexpected in a subduction-related accretionary prism.

**Keywords**

Pulo do Lobo metapelites

Low-pressure gradient

X-Ray diffraction

Chlorite geothermometry

Raman spectroscopy of carbonaceous material

**Highlights**

A multidisciplinary approach has been applied to study the metamorphism of the Pulo do Lobo metapelites.

Devonian metamorphism entered epizone conditions.

Carboniferous metamorphism reached the anchizone/epizone boundary.

The inferred low-pressure gradient is incompatible with a subduction-related accretionary prism.



## 1. Introduction

The knowledge of temperature and pressure conditions reached by the low-grade
metasedimentary units stacked hinterlands of orogens helps to better interpret their
tectonometamorphic evolution (e.g., Goffé and Velde, 1984; Franceschelli et al., 1986; Ernst,
1988; Gutiérrez-Alonso and Nieto, 1996; Frey and Robinson, 1999; Bousquet et al., 2008;
Lanari et al., 2012). In this regard, the various results derived from the application of diverse
geothermometric and/or geobarometric methods may also allow the identification and
characterization of superposed tectonometamorphic events, thus improving the knowledge
of the P-T paths and their tectonic significance (e.g., Brown, 1993; Crouzet et al., 2007; Ali,
2010; Lanari et al., 2012; Airaghi et al., 2017).
The metamorphism of the Iberian Variscides has been mostly studied on intensely
metamorphosed rocks in order to characterize high-grade events and obtain the P-T-t paths
of suture-related units (e.g., Gil Ibarguchi et al., 1990; Abalos et al., 1991; Escuder Viruete et
al., 1994; Barbero, 1995; Arenas et al., 1997; Fonseca et al., 1999; López-Carmona et al., 2013;
Martínez Catalán et al., 2014). The low- to very low-grade units have been also studied (e.g.,
Martínez Catalán, 1985; Bastida et al., 1986, 2002; López Munguira et al., 1991; Gutiérrez-
Alonso and Nieto, 1996; Abad et al., 2001, 2002, 2003a; Martínez Poyatos et al., 2001; Nieto
et al., 2005; Vázquez et al., 2007), despite the scarcity of appropriate robust methodologies
to apply in these kind of rocks. Obtaining new results from the low-grade rocks of the Pulo
do Lobo belt, a suture-related low-grade unit in SW Iberia, is of capital importance in order
to understand its significance and tectonometamorphic evolution, that have been cause of
discrepancies, and to reconstruct the overall history of the SW Iberian Variscides.
In this work, three different methodologies are applied to a number of samples of the Pulo
do Lobo belt (Fig. 1): (i) X-Ray Diffraction (XRD) in order to identify minerals not easily
recognizable with optical microscopy (fine-grained muscovite, paragonite, mixed-layer
phyllosilicates, etc.) and obtain thermobarometric information via the measurement of
crystallographic parameters (illite "crystallinity" and $b$-cell dimension); (ii) Compositional
maps derived from electron probe microanalysis (EPMA), which enable the recognition of
different tectonometamorphic events by combining mineral composition and microtextural
features (e.g., Airaghi et al., 2017), as well as the application of geothermobarometers based
on chlorite and K-white mica compositions; and (iii) Raman spectroscopy of carbonaceous
material (RSCM) to estimate peak temperatures thanks to an adapted thermometric
calibration. Firstly, the results obtained enables discussing the tectonometamorphic
evolution of the Pulo do Belt. Moreover, the comparison of the different approaches allows
know further their reliability and sensitivity to characterize different geological processes.

## 2. Geological setting

The SW Iberian Variscides resulted from the Devonian-Carboniferous left-lateral oblique
collision of three different terranes: the Central Iberian Zone (CIZ), the Ossa-Morena Zone
(OMZ) and the South Portuguese Zone (SPZ) (Fig. 1a). The boundaries between these
terranes are considered as orogenic sutures (Pérez-Cáceres et al., 2016, and references
therein). Besides the dominant left-lateral shortening kinematics, SW Iberia also attests



Mississippian synorogenic sedimentary basins, widespread mafic magmatism and high-
temperature metamorphic areas, which altogether reveal an intraorogenic transtensional
stage (Simancas et al., 2003, 2006; Pereira et al., 2012; Azor et al., 2019).
The OMZ is commonly interpreted as a continental piece that drifted from the CIZ (i.e.,
north Gondwana) in early Paleozoic times (Matte, 2001). The OMZ/CIZ suture (Badajoz-
Córdoba Shear Zone) includes early Paleozoic amphibolites with oceanic affinity, eclogite
relics and intense high- to low-grade left-lateral shear imprint (Burg et al., 1981; Abalos et
al., 1991; Azor et al., 1994; Ordóñez-Casado, 1998; López Sánchez-Vizcaíno et al., 2003;
Pereira et al., 2010). Ediacaran to Carboniferous sedimentary successions with an
unconformity at the base or the Lower Carboniferous characterize the OMZ. Low-grade
regional metamorphism dominates the OMZ, though there are areas of high-temperature /
low-pressure metamorphism associated with Early Carboniferous magmatism (e.g. Bard,
1977; Crespo-Blanc, 1991; Díaz Azpiroz et al., 2006; Pereira et al., 2009).
The SPZ is a continental piece considered as a fragment of Avalonia (Pérez-Cáceres et al.,
2017 and references therein). Thus, the OMZ/SPZ boundary is usually interpreted as the
Rheic Ocean suture (Pérez-Cáceres et al., 2015 and references therein). This boundary is
delineated by the Beja-Acebuches Amphibolites (Fig. 1b), a narrow strip of metamafic rocks
that resembles a dismembered ophiolitic succession (from greenschists to metagabbros and
locally ultramafic rocks) (e.g., Bard, 1977; Crespo-Blanc, 1991; Quesada et al., 1994). This
unit was interpreted as a Rheic ophiolite (Munhá et al., 1986; Crespo-Blanc, 1991; Fonseca
and Ribeiro, 1993; Quesada et al., 1994; Castro et al., 1996), though this idea was reconsidered
based on the Mississippian age of the mafic protoliths (≈340 Ma; Azor et al., 2008). Actually,
the Beja-Acebuches unit is better interpreted as an outstanding evidence of the early
Carboniferous intraorogenic, lithospheric-scale transtensional and magmatic episode that
here obscures the previous suture-related features of the OMZ/SPZ boundary (Pérez-
Cáceres et al., 2015 and references therein). The rocks of the Beja-Acebuches Amphibolites
were affected by a left-lateral ductile shearing occurred at granulite to greenschist facies
conditions, though amphibolite facies conditions were dominant (e.g., Quesada et al., 1994;
Castro et al., 1996; Castro et al., 1999; Díaz Azpiroz et al., 2006). This metamorphism has
been dated at 345-330 Ma (Dallmeyer et al., 1993; Castro et al., 1999), thus suggesting that it
started very shortly after the magmatic emplacement.
North of the Beja-Acebuches Amphibolites, the allochthonous Cubito-Moura unit might be
the only witness of the Rheic Ocean suture (Fonseca et al., 1999; Araújo et al., 2005; Pérez-
Cáceres et al., 2015). This unit was emplaced onto the southern OMZ border (Fig. 1b) with
a left-lateral top-to-the-ENE kinematics (Ponce et al., 2012). It contains Ediacaran-Lower
Paleozoic metasediments and Ordovician MORB-featured mafic rocks (≈480 Ma; Pedro et
al., 2010) transformed into high-pressure blueschists and eclogites at ≈370 Ma (Moita et al.,
2005). The high-pressure metamorphism has also been studied by using white mica and
chlorite (and chloritoid pseudomorphs) mineral equilibria (Booth-Rea et al., 2006; Ponce et
al., 2012; Rubio Pascual et al., 2013), yielding peak conditions of 1 GPa at 450 °C.
South of the Beja-Acebuches Amphibolites, low- to very low-grade successions crop out in
the SPZ: Devonian siliciclastics, earliest Carboniferous volcano-sedimentary rocks, and a
south-migrating Carboniferous flysch (e.g., Oliveira, 1990). The SPZ can be divided, from





north to south, into the Pulo do Lobo belt (see below), the Iberian Pyrite belt (that includes
massive sulphide deposits) and the Carboniferous flysch. The deformation in the SPZ
consists in a south- to southwest-vergent fold and thrust belt with decreasing strain intensity
and age southwards (Oliveira, 1990; Simancas et al., 2004). The metamorphic grade also
decreases southwards, from epizone to diagenesis, through the SPZ (Munhá, 1990; Abad et
al., 2001).

## 2.1. Pulo do Lobo belt

The northernmost unit of the SPZ is the Pulo do Lobo belt, whose evolution is intimately
related to the OMZ/SPZ suture (Fig. 1b). The Pulo do Lobo belt constitutes a polydeformed
structure affecting low-grade Devonian-Carboniferous sedimentary formations. These
formations are, from bottom to top (Fig. 1b-c):
(i) The Pulo do Lobo formation (s. str.) is constituted by a succession of satiny black to grey
phyllites and fine-grained schists with minor intercalations of quartz sandstones (Fig. 2a).
The presence of abundant segregated quartz veins (pre- to post-folding) is common. The
palynological content suggests a middle Frasnian age (Pereira et al., 2018).
(ii) The Ribeira de Limas formation is constituted by phyllites with thin beds of quartz
sandstones and arkoses (Fig. 2b). The presence of palynomorphs also suggests a middle
Frasnian age for this formation (Pereira et al., 2018). The contact with the underlying Pulo
do Lobo formation is gradual, with a progressive increase of sandstones and a decrease of
phyllites upwards. For that reason, we will refer to the Pulo do Lobo and Ribeira de Limas
formations as the lower formations of the Pulo do Lobo belt. Furthermore, these lower
formations share the same structure consisting in three fold-related foliations (Fig. 2a-b;
Pérez-Cáceres et al., 2015). The first foliation of the lower formations ($S_1$) is preserved inside
microlithons of the second foliation ($S_2$); usually, the angle between these two foliations is
high. $S_2$ is the main foliation and consists in a crenulation-dissolution cleavage that frequently
appears as a milimetric- to centimetric-spaced tectonic banding. This foliation is axial-plane
to north-vergent folds. The third foliation ($S_3$) is a spaced crenulation-dissolution cleavage
that sometimes develops a characteristic decimetric- to metric-scale tectonic banding. $S_3$ is
associated with upright to slightly south-vergent folds.
(iii) The Santa Iría formation is composed by alternating beds of slates and greywackes (Fig.
2c). The greywacke beds show normal grading and erosive base. Paleontological and
palynostratigraphic studies suggest an Upper Famennian age for this formation (Pereira et
al., 2008; 2018). However, an early Carboniferous age is much plausible, since more than
90% of the palynomorphs correspond to reworked material (Lopes et al., 2014) and the
younger detrital zircon population is early Carboniferous (Braid et al., 2011; Pérez-Cáceres
et al., 2017; Pereira et al., 2019). The Santa Iría formation only shows two foliations,
correlative with the last two deformation phases in the lower formations. Therefore, an
unconformity between them is inferred, which also agrees with the age and flysch character
of the Santa Iría formation (Pérez-Cáceres et al., 2015). $S_2$ is observed as a penetrative slaty
cleavage, while $S_3$ is a disjunctive crenulation cleavage.





According to the evolutionary model proposed by Pérez-Cáceres et al. (2015), the two main
foliations ($S_2$ and $S_3$) in the Pulo do Lobo belt resulted from the middle/upper Carboniferous
collision between the OMZ and SPZ. On the contrary, the first foliation ($S_1$) in the Pulo do
Lobo belt might have formed during the vanishing stages of Rheic Ocean subduction and/or
the starting Variscan collision, probably at Late Devonian time.
The Pulo do Lobo belt contains some decimetric- to metric-scale lenticular bodies of
MORB-featured metamafic rocks intercalated within the phyllites of the Pulo do Lobo
formation and interpreted as a tectonic mélange (the so-called Peramora Mélange; Fig. 1b-c;
Apalategui et al., 1983; Eden, 1991; Dahn et al., 2014). Based on this aspect and on the
supposedly Rheic Ocean derived greenschists, the Pulo do Lobo belt has been classically
interpreted as a pre-collisional subduction-related accretionary prism (Eden and Andrews,
1990; Silva et al., 1990; Eden, 1991; Braid et al., 2010; Ribeiro et al., 2010; Dahn et al., 2014).
However, the recently obtained Mississippian U/Pb zircon ages from the metamafic rocks
(Dahn et al., 2014; Pérez-Cáceres et al., 2015) make difficult to maintain such hypothesis.
More properly, they can be interpreted as mafic intrusions/extrusions in the frame of the
intraorogenic transtensional magmatic event that prevailed in SW Iberia during the
Mississippian. The metamafic rocks display a foliation (equivalent to the $S_2$ of the enveloping
metasediments) developed at loosely constrained greenschist facies conditions. These rocks
would have been imbricated with the Pulo do Lobo metasediments during the second
deformation phase which caused $S_2$ (Peramora Olistostrome; Pérez-Cáceres et al., 2015). Our
multidisciplinary metamorphic study of the Pulo do Lobo metasediments provides with
crucial data concerning the tectonic significance of this belt.

**3. Samples and analytical methods**
Eighteen samples were collected from well-exposed outcrops of phyllosilicate-rich detrital
rocks of the Pulo do Lobo belt along two north-south transects perpendicular to the
structural trend. Five samples belong to the Santa Iría formation (unconformable upper
formation) and thirteen to the lower formations (location of samples are in the map and
cross-sections of Fig. 1b-c and the UTM coordinates in supplementary information). As a
whole, the samples were selected in not altered outcrops, far from faults and joints, and were
taken as homogeneous as possible. Sampling design was intended to collect representative
sites, both of the overall stratigraphic succession and along the two transects. We also aimed
to characterize the unconformity between the lower and upper formations from a
metamorphic point of view, since "crystallinity" aspect at first sight seems to be lower in the
Santa Iría formation. Some samples from the lowermost Pulo do Lobo formation were
collected not far from the metabasite lenses of the Peramora Mélange.
Samples were examined under the optical microscope and SEM for overall mineralogy,
deformation and minerals/foliations relationships using an environmental scanning electron
microscope FEI model Quanta 400, operating at 15–20 keV (Centro de Instrumentación
Científica-CIC, University of Granada, Spain).






### 3.1. X-Ray diffraction

Sample preparation and analysis by XRD were done in the laboratories of the Department of Mineralogy and Petrology of the University of Granada (Spain). After washing and cleaning of patinas and oxides, samples were crushed to a <2 mm fraction. The <2 µm fractions were separated by repeated extraction of supernatant liquid after centrifugation, according to the Stokes' law. Oriented aggregates were prepared by sedimentation on glass slides of whole-rock and <2 µm fractions (the latter aims to minimize the content of detrital micas non-re-equilibrated during very low-grade metamorphism, which are generally larger than 2 µm; Moore and Reynolds, 1997). Samples were also treated with ethylene glycol (EGC) to identify illite/smectite or chlorite/smectite mixed-layers on the basis of their expansibility. Samples were analyzed using a PANalytical X'Pert Pro powder diffractometer equipped with an X'Celerator detector, CuKα radiation, operated at 45 kV and 40mA, Ni filter and 0.25° divergence slit. The resulting diffraction diagrams were examined to extract information on mineralogy based on their characteristic reflections and white mica crystal data.

The Illite "Crystallinity" index (Kübler Index; KI; Kübler, 1968) has been estimated from the measurement of the full peak-width of K-white mica at half maximum intensity (FWHM values), expressed as $\Delta°2\theta$ of the Bragg angle. Preparation of samples and experimental conditions were carried out according to IGCP 294 IC Working Group recommendations (Kisch, 1991). A step increment of 0.008° $2\theta$ and a counting time of 52 s/step were used in the diffractometer. The KI has been measured in all samples for both the 5 and 10 Å reflection peaks of K-white mica in order to identify possible effects of other overlapping phases (Nieto and Sánchez-Navas, 1994; Battaglia et al., 2004). Some XRD traces showing complex mixture of mixed-layered minerals were decomposed with the MacDiff software (Petschick, 2004). The FWHM values obtained in the laboratory (x) have been transformed to Crystallinity Index Standard (CIS) values (y) using the equation y=0.972x + 0.1096 (R2 = 0.942), obtained from the measure in our lab of the international standards of Warr and Rice (1994). Finally, they have been expressed in term of traditional KI values using the equation of Warr and Ferreiro Mähnlmann (2015; 'CIS' = 1.1523*Kübler index 'Basel lab' + 0.036). The lower and upper boundaries of the anchizone in the KI scale are 0.42 and 0.25 °2θ, respectively (Warr and Ferreiro Mähnlmann, 2015). The thermal range for the anchizone is estimated in c. 200-300 °C, though the KI cannot be considered as a true geothermometer (Frey, 1987; Kisch, 1987).

The $b$-cell parameter of white mica was obtained from the (060) reflection peak measured with quartz as internal standard on polished rock-slices cut normal to the sample main foliation (Sassi and Scolari, 1974). The $b$-cell dimension of K-white mica is often proportional to the magnitude of phengitic substitution and therefore considered as a proxy of the pressure conditions during its crystallization. Thus, Guidotti and Sassi (1986) have shown that $b$ values lower than 9.000 Å are typical of low-pressure facies conditions, while $b$ values higher than 9.040 Å are related to rather high-pressure facies metamorphism. Precise measurements of the basal spacing of white mica ($d_{001}$) have also been made, using quartz from the sample itself as internal standard. $d_{001}$ is related to the paragonitic Na/K substitution



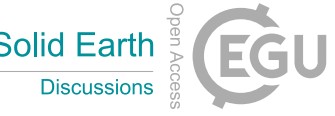

(Guidotti et al., 1992), thereby approximately reflecting the temperature of formation
(Guidotti et al., 1994).

### 3.2. EPMA-derived X-Ray compositional maps and chlorite thermometry

From all of the collected samples, we selected those with the larger phyllosilicate grain-size
for electron probe microanalysis (EPMA). Thus, three carbon-coated polished thin-sections
were studied. The selected samples (PLB-84, PLB-88 and PLB-93) belong to the lower
formations of the Pulo do Lobo belt (Fig. 2d-e). The Santa Iría samples could not be studied
due to the tiny grain size of the slaty minerals (commonly less than 3 μm).
Compositional maps and accurate spot analyses were performed on a JEOL JXA-8230
EPMA at the Institut des Sciences de la Terre (ISTerre) in Grenoble (France), according to
the analytical procedure proposed by de Andrade et al. (2006) and Lanari et al. (2014a). The
data acquisition was made in wavelength dispersive spectrometry mode (WDS). Ten
elements (Si, Ca, Al, K, Mn, Na, P, Ti, Fe and Mg) were analyzed using five WD
spectrometers: TAP crystal for Si and Al, PETL for Ti and P, TAPH for Na and Mg, PETH
for K and Ca, and LIFH for Mn and Fe. The standardization was made by using certified
natural minerals and synthetic oxides: Wollastonite (Si, Ca), Corundum (Al), Orthoclase (K),
Rhodonite (Mn), Albite (Na), Apatite (P), Rutile (Ti), Hematite (Fe), and Periclase (Mg). X-
Ray maps were obtained by adding successive adjacent profiles. Beam current of 100 nA and
beam size spot (focused) were used. The step (pixel) size was 1 μm and dwell time was 200-
300 msec per pixel. Spot analyses were obtained along the profiles within the mapping at 15
kV accelerating voltage, 12 nA beam current and 2 μm beam size spot (focused). The on-
peak counting time was 30 sec for each element and 30 sec for two background
measurements at both sides of the peak. ZAF correction procedure was applied. The internal
standards were orthoclase and/or chromium-augite (Jarosewich et al., 1980), which were run
(3 points on each standard) after each profile in order to monitor instrumental drift and
estimate analytical accuracy. Drift correction was made, if necessary, using the corresponding
regression equation.
The WDS X-Ray maps were then processed with XMapTools
(http://www.xmaptools.com), a MATLAB©-based graphical user interface program to
process the chemical maps, link them to thermobarometric models and estimate the
pressure-temperature conditions of crystallization of minerals in metamorphic rocks (Lanari
et al., 2014a). The compositional maps were standardized with the spot analyses measured
along the profiles and mineral compositions were plotted into binary and ternary diagrams
using the interface modules *Chem2D and Triplot3*D. Chemical maps of amount of tetrahedral
aluminum (Al$^{IV}$) of chlorites were acquired, because is at the base of many empirical chlorite
thermometers (e.g. Cathelineau and Nieva, 1985; Cathelineau, 1988). The temperature
conditions were estimated for each chlorite pixel of the maps using the chlorite thermometer
of Lanari et al. (2014b), as well as the approaches of Vidal et al. (2006) and Bourdelle et al.
(2013), which are summarized in the supplementary information.
In addition to the above mentioned compositional maps, white micas from seven carbon-
coated thin sections of the lower formations of the Pulo do Lobo belt were analyzed before



with a Jeol four-spectrometer microprobe (JXA-8200 Superprobe) at the University of
Huelva (Spain). A combination of silicates and oxides were used for calibration. Single point
analyses were obtained with 10 nA probe current, 1-5 μm spot size, and 20 kV of acceleration
voltage.

### 3.3. Raman Spectroscopy of carbonaceous material

Beyssac et al. (2002a) calibrated a technique for the quantification of peak metamorphic
temperature, which can be used even in the absence of specific mineral assemblages
necessary for classical thermobarometric estimates. This technique, Raman Spectroscopy of
Carbonaceous Material (RSCM), is based on the observation that sedimentary carbonaceous
material is progressively transformed into graphite at increasing temperature. Beyssac et al.
(2002a) found a linear relationship between temperature and the structural state of CM
quantified by Raman microspectroscopy. Because of the irreversible character of
graphitization, CM structure is not sensitive to the retrograde path during exhumation of
rocks, but only depends on the maximum temperature reached during metamorphism
(Beyssac et al., 2002a). Temperature can be determined in the range 330-650°C with a
calibration-attached accuracy of ± 50 °C due to uncertainties on petrologic data used for the
calibration. Relative uncertainties on temperature are, however, much smaller (around 10-15
°C; Beyssac et al., 2004). For temperature below 330 °C, Lahfid et al. (2010) performed a
systematic study of the evolution of the Raman spectrum of CM in low-grade metamorphic
rocks in the Glarus Alps (Switzerland). They showed that the Raman spectrum of CM is
slightly different from the spectrum observed at higher temperature and they established a
quantitative correlation between the degree of ordering of CM and temperature.
In this work, twelve representative thin-sections previously examined by optical microscopy
were selected. From them, ten samples were finally analyzed: eight samples belong to the
lower formations (Pulo do Lobo and Ribeira de Limas formations), while the other two
belong to the Santa Iría formation. Polished thin-sections cut perpendicularly to the foliation
were analyzed at the Institut de Minéralogie, de Physique des Matériaux et de Cosmochimie
at the Sorbonne University of Paris (France). We followed closely the analytical procedure
described by Beyssac et al. (2002a, b; 2003; see supplementary information). More than 15
Raman spectra (Fig. 3) were obtained for each sample using a Renishaw InVIA Reflex
microspectrometer equiped with a 514.5 nm Modulaser argon laser under circular
polarization. The laser was focused by a DMLM Leica microscope, and laser power was set
below 1 mW at the sample surface. The Rayleigh diffusion was eliminated by edge filters and
the signal was dispersed using a 1800 g/mm grating and finally analyzed by a Peltier cooled
RENCAM CCD detector. The recorded spectral window was large to correctly set the
background correction, from 700 to 2000 cm$^{-1}$ in case of low-temperature samples. Before
each session, the spectrometer was calibrated with a silicon standard. CM was systematically
analyzed behind a transparent adjacent mineral, generally quartz or white mica grains oriented
along $S_1$. For a full description of the temperature calculations see the supplementary
information.





## 4. Results


Acording to SEM analysis, all the samples correspond to slates or phyllites with
phyllosilicates smaller than 500 µm, composed of variable quartz + K-white mica ± chlorite
± feldspar ± ore and accessory minerals (Fig. 2d-f). Samples from the Santa Iría formation
have much smaller grain-size and apparently lower "crystallinity" (Fig. 2f). The first foliation
$S_1$ is defined by the largest micas and chlorites (Fig. 2d-e), being folded by microscopic- to
centimetric-scale tight folds of the second deformation phase (Fig. 2a-b, d-e). The second
foliation $S_2$ is the main foliation at outcrop (Fig. 2a-c), but the development of phyllosilicates
(mostly white mica) is lesser than $S_2$. The third foliation $S_3$ is much less penetrative (Fig. 2a-
c) and does not develop phyllosilicates. Large detrital phyllosilicate clasts have not been
observed.

## 4.1. X-Ray diffraction


The mineralogy and crystal parameters obtained from the 18 samples of the Pulo do Lobo
belt are summarized in Table 1. The results of KI values, b-cell parameter and $d_{001}$ analyzed
in K-white mica, obtained from whole-rock and <2 µm fractions are very similar, which
suggests that detrital micas re-equilibrated during metamorphism.
The mineralogy of the samples is relatively simple: Qz + Ms + Fsp+ Chl ± Pg ± C/S. The
slates of the Santa Iría formation have quartz, muscovite and chlorite, with chlorite/smectite
interlayers (C/S) in some samples. In the lower formations, besides quartz and muscovite,
chlorite is present in almost all of the samples, paragonite appears in most of them, and
chlorite/smectite interlayers are occasional.
KI values measured in the 10 Å peak of white mica from the <2 µm fraction are shown in
Table 1 and Fig. 1c with a relative colour bar from orange (lower values) to green (higher
values). Values of the Santa Iría samples (n=5) range from 0.20 to 0.26 Δ°2θ, the mean value
being 0.23 (standard deviation 0.02). As for the lower formations (n=12), KI values range
from 0.17 to 0.22, the mean value being 0.19 (standard deviation 0.02). KI values measured
in the 5 Å peak (not shown) are very similar to those of the 10 Å peak.
The measured $b$-cell parameter of white mica varies in a close range around 9 Å (8.991-9.002).
Mean value is 8.995 Å (standard deviation 0.003) for the Santa Iría formation samples, and
8.997 Å (standard deviation 0.003) for the samples of the lower formations. $d_{001}$ values
average 9.992 Å (standard deviation 0.004) and differ slightly between upper and lower
formations, being higher in the upper formation.
The results obtained through X-Ray diffraction denote very low- to low-grade metamorphic
conditions due to the presence of C/S and KI values between 0.17-0.26 Δ°2θ. In addition,
$b$-cell parameters show a low-pressure metamorphic gradient.

## 4.2. Compositional maps and chlorite thermometry


X-Ray maps show the distribution of major elements and allow identifying white mica,
chlorite, and some albite porphyroblasts, with ilmenite and rutile as accessory minerals (Fig.



4a-b). Although quartz is abundant in all of the samples, the zoomed selected areas for X-ray mapping (composed mostly by phyllosilicates) do not contain quartz (Fig. 4a-b). White mica is abundant along both $S_1$ and $S_2$ foliations (Fig. 2d-e and 4b). Chlorite is found mostly along $S_1$, being very scarce and small-sized along $S_2$ (Fig. 2e and 4b), with the exception of sample PLB-93 where chlorite is similar in amount in both foliation domains (Fig. 4b).

Mapped compositions of end-members of white mica and chlorite have been plotted in the ternary diagrams of Figure 5. The composition of white mica is similar in the three maps. It is close to muscovite, with 25% of pyrophyllite and very scarce celadonite content (Fig. 5a). The high content of pyrophyllite (high amount of interlayer vacancies) is typical of low-pressure illite compositions. Figure 6 shows white mica compositional ratios, which can be related to P/T conditions: they present low degree of Na substitution and low phengitic component, thus being close to the muscovite end-member. These results point to low-pressure conditions and agree well with XRD results: low $b$-cell parameter and high $d_{001}$ (Table 1).

Chlorite compositions are variable, though all of them have in common ≈50% clinochlore + daphnite and ≈50% amesite + sudoite (Fig. 5b). Chlorites in sample PLB-88 are poor in amesite with a large variation of clinochlore + daphnite and sudoite. In sample PLB-84 chlorites, variable compositions between amesite and sudoite indicate a variation of $Al^{IV}$, which implies an increase of temperature from rims to cores as shown in the chemical maps of Fig. 4c. Finally, PLB-93 chlorites are poor in sudoite, thus suggesting higher average temperatures. Altogether, chlorite compositional data suggest the presence of two end-members: sudoite-rich low-temperature (PLB-88), and amesite-rich high-temperature (PLB-93).

Maps of $Al^{IV}$ in chlorites have been represented in Fig. 4c. Sample PLB-88 shows lower $Al^{IV}$ content (≈1.1-1.3 apfu) than sample PLB-93 (≈1.3-1.5 apfu). In sample PLB-84, some large chlorite grains oriented along $S_1$ are zoned, with higher $Al^{IV}$ content in the cores (≈1.4 apfu) than in the rims (≈1.0 apfu; see white square in Fig. 4c). According to the empirical calibration of Cathelineau (1988), $Al^{IV}$ in chlorites increases with temperature. Thus, the $Al^{IV}$ content in chlorites manifests different temperatures in different samples, and also from core to rim in singular grains.

Temperature maps have been obtained with the semi-empirical thermometer of Lanari et al. (2014b), assuming that $Fe^{2+}$ is the Fe total (Fig. 4d). Temperatures range between 100-200 °C in sample PLB-88, 150-350 °C in sample PLB-84, and 200-450 °C in sample PLB-93. Tiny chlorites developed along $S_2$ show lower temperatures than larger and more abundant chlorites along $S_1$, with the exception of sample PLB-93. Furthermore, some large chlorites oriented along $S_1$ are zoned, showing high-temperature relic cores (350-450 °C; see white insets in Fig. 4c-d) surrounded by low-temperature rims (150-250 °C).

To test Vidal et al. (2005, 2006) and Bourdelle et al. (2013) approaches, an area of representative chlorites in an $S_1$ microlithon was selected from each map (see red insets in Fig. 4d). Corresponding chlorite compositions were extracted and introduced in the chlorite-quartz-water equilibria (Fig. 7a, Vidal et al., 2005, 2006; Fig. 7b, Bourdelle et al., 2013). The temperature estimates (Fig. 7) are fairly similar with both methods, averaging 120-230 °C in sample PLB-88 and 150-380 °C in sample PLB-84. This is also in agreement with the





temperature maps calculated with the Lanari et al. (2014a) model. Only the sample PLB-93
shows a divergence on temperature averages: mostly 200-250 °C with the thermometer of
Bourdelle et al. (2013), and 250-350 °C with the one of Vidal et al. (2005, 2006). Nevertheless,
the Bourdelle thermometer predicts temperatures up to 380-400°C. In both cases, the higher
temperature analyses are obtained from crystal cores and belong to the sample PLB-93.

### 4.3. RSCM thermometry

The ratio parameters and corresponding maximum temperatures obtained from all the
spectra analyzed are shown in the supplementary information. The Raman spectra were
decomposed into bands following the appropriate fitting procedure described in Beyssac et
al. (2002a) for the lower formations (high-temperature Raman spectra; ratio parameter R2)
and Lahfid et al. (2010) for the Santa Iría formation (low-temperature Raman spectra; ratio
parameters RA1 and RA2). The average temperatures are shown in Table 1 and Fig. 1c with
a relative colour bar from red (higher temperature) to blue (lower temperature). The average
temperatures for the lower formations range from 420 to 530 °C, with a mean value of 468
°C (standard deviation of 35). The highest temperatures are found in samples PLB-82 (530
°C) and PLB-93 (495 °C), while the remaining ones do not exceed 480 °C. As for the Santa
Iría formation, temperatures are lower (315-330 °C; Table 1) than in the underlying
formations.

### 5. Interpretation and discussion

### 5.1. Deformation/metamorphism relationships

The obtained analytical results must be interpreted in the context of the Variscan evolution
of the Pulo do Lobo belt. As described above, two regional deformational events $D_1$ and $D_2$
gave way to the development of foliations (Devonian $S_1$ and Carboniferous $S_2$) accompanied
by metamorphic phyllosilicate growth ($M_1$ and $M_2$). In the cross-sections of Fig. 1c, KI values
derived from XRD and average temperature from RSCM are represented. The lowest
metamorphic grade (green and blue colours) corresponds to the Santa Iría formation.
Moreover, Table 2 summarizes the relationship between deformation and metamorphism of
the Pulo do Lobo belt in the context of the Variscan tectonic evolution of SW Iberia (Pérez-
Cáceres et al., 2015).
The textural observations evidence that in most samples of the lower formations $M_1$ was the
main crystallization event, developing abundant and large-sized white mica and chlorite in $S_1$
microlithons, while $M_2$ gave way to small-sized white mica (e.g., Fig. 2e and map 1 in Fig. 4).
On the other hand, polydeformed rocks commonly show previously grown minerals rotated
towards a new foliation developed at lower-grade conditions, without new crystallization.
This can be the case of the white micas that define $S_2$ in some samples (illustrated in Fig. 2d),
which, in turn, is not contradictory with the similar chemical composition of $S_1$ and $S_2$ micas
(Fig. 5a). As shown in our samples, $S_1$ is variably crenulated by $D_2$, so that $M_1$ minerals are
variably rotated towards $S_2$. Consequently, the metamorphic data obtained from the samples
of the lower formations will be ascribed to $D_1$-$M_1$. Sample PLB-93 might represent an
exception, since its slightly higher RSCM and chlorite-derived temperatures might be due to





nearby intrusions (Fig. 1b and 1c.1). At this respect, it is important to note the Mississippian
transtensional event (basins development and abundant mafic magmatism) that took place
between $D_1$ and $D_2$ (Pérez-Cáceres et al., 2015). The characterization of $M_2$ can be done by
studying the samples from the Santa Iría formation, which are only affected by $S_2$
accompanied by small-sized phyllosilicate growth (Fig. 2f).

**5.2. First tectonothermal event (Devonian $M_1$)**
The observed mineral association (Qz + Ab + Ms + Chl ± Pg), together with the presence
of C/S is compatible with low-grade metamorphic conditions (Table 1). White mica
"crystallinity" values (0.17-0.22 Δ°2θ; average 0.19) are always in the range of the epizone
(low-grade or greenschists facies; >300 °C; Frey, 1987; Kisch, 1987, Warr and Ferreiro
Mähnlmann, 2015), in accordance with the values reported by Abad et al. (2001) in a more
general study of the diagenetic-metamorphic evolution of the South Portuguese Zone
metapelites. Nevertheless, both the values of KI, still far from 0.14 Δ°2θ, and their variability,
suggest that temperature was not high enough as to stabilize a highly crystalline white mica
at high epizone conditions (Abad et al., 2006). This is in agreement with the low Na content
of K-micas coexisting with paragonite (Fig. 6), meaning a very-low temperature position in
the muscovite-paragonite solvus for natural quasi-binary Pg-Ms pairs (Guidotti et al., 1994).
By contrast, the maximum temperatures obtained with RSCM geothermometry are
surprisingly high (420-530 °C; average 470 °C; corresponding to very high epizone or even
medium-grade conditions; Table 1).
The composition of paired chlorite and white mica is normally used to calculate pressure and
temperature (e.g., Vidal et al., 2006), but multi-equilibrium approach was not successful
because the P-T equilibrium conditions did not converge. This result is indicative of chemical
disequilibrium, precluding their use as a reliable geothermobarometer in this case. The
temperatures calculated from chlorite compositions following various approaches (Vidal et
al., 2006, Fig. 7a; Bourdelle et al., 2013, Fig. 7b; Lanari et al., 2014a, Fig. 4d) are as follow:
120-230 °C for sample PLB-88, 150-380 °C for sample PLB-84, and 250-400 °C for sample
PLB-93 and a small population of chlorite cores from sample PLB-84 (Figs. 4 and 7, and
Table 1). The slightly higher temperature of sample PLB-93 is inferred from its highest white
mica "crystallinity" (0.17 Δ°2θ), high RSCM temperature (495 °C), high-temperature
(amesite-rich) chlorite and higher chlorite thermometry (Table 1), and can be explained by
its nearness to metric-scale mafic igneous bodies of the Peramora Mélange (located at ≈200
m to the south; Pérez-Cáceres et al., 2015) and/or to a granite stock (located at ≈5 km to the
west) (Fig. 1b).
In our samples there is some evidence of chlorite retrogression: (i) the chemical
disequilibrium showed by the white mica/chlorite geothermobarometer, (ii) the presence of
C/S mixed layers not stable in the epizone (e.g. Potel et al., 2006), (iii) the difference between
temperature estimates from crystal rims to cores, and the higher temperature relic cores
preserved in large chlorites defining $S_1$ (Fig. 4c-d), and (iv) the previously reported XRD and
TEM data of chlorite retrograded to smectite and corrensite in the Pulo do Lobo belt (see
fig. 1 in Nieto et al., 2005). The existence of chlorites with different compositions crystallized
at different temperatures is the usual scenario (e.g., Vidal et al., 2006, 2016; Lanari et al., 2012;





2014a and b; Grosch et al., 2012; 2014; Cantarero et al., 2014). In such situation, the
definition of a single temperature and pressure attributable to peak conditions is difficult.
The maximum temperature showed by chlorite relic cores is 350-450 °C (Fig. 4d), which is
more in accordance with the conditions estimated for $M_1$ by means of white mica
"crystallinity" and RSCM data.
An issue that deserves some discussion is the difference in temperature estimates between
RSCM and other techniques. RSCM thermometry records the peak temperature and is not
sensitive to the retrograde path. Alternatively, other methods based on phyllosilicate
compositions are prone to record reequilibration during the retrograde path; thus, they rarely
record the peak conditions, except perhaps in the core of certain large crystals. Therefore,
RSCM and phyllosilicate-based methods do not record the same information on
temperature, being in fact complementary. The analyzed CM gains were carefully checked by
microtextural observation and spectral geometry to make sure that these grains are actually
derived from in situ organic matter graphitized during metamorphism.
In our case study, at the high peak temperature given by the RSCM thermometry, minerals
such as biotite or garnet are expected to crystallize in metasediments, though they have not
been observed in our samples. The absence of such minerals can be due to whole-rock
composition, and explained by growth inhibition related to Na-excess, as evidenced by the
presence of albite and paragonite in our samples. Another possible explanation could be the
higher sensitivity of CM graphitization to fast reequilibration during a short-time thermal
event. Thus, the Mississippian intrusions subsequent to $M_1$ in the Pulo do Lobo formation
(see description in section 2) could have exerted a fast and locally intense thermal imprint
that influenced CM but not the crystal chemistry of silicates. Moreover, recrystallization
processes are not only function of temperature, but also promoted by deformation/stress,
time, fluid/rock ratio (Merriman and Frey, 1999). Observations of this kind (differing
reaction kinetics between organic and inorganic material (e.g. illite) in a contact metamorphic
setting can be found in Olsson (1999) and Abad et al. (2014). Regarding the time of geological
processes, Mori et al. (2017) investigated the importance of heating duration for RSCM
thermometry by studying graphitization around dykes. They showed that small-scale
intrusions generating short thermal events modify the structure of CM in the surrounding
rocks, to conclude that CM crystallinity is clearly related to contact metamorphism. The
influence of low-pressure contact aureoles on RSCM temperature patterns is further
supported by the results obtained by Hilchie and Jamieson (2014), who concluded that the
variation of RSCM temperatures can be controlled by the subsurface geometry of a pluton.
Finally, the long-distance thermal influence of plutonic intrusions on low-grade rocks located
as far as 10 km has already been evidenced (e.g., Merriman and Frey, 1999; Martínez Poyatos
et al., 2001) and could also be recorded by RSCM thermometry in our samples.

### 566    5.3. Second tectonothermal event (middle/upper Carboniferous $M_2$)

The mineralogy of the Santa Iría samples (Qz + Ms + Chl ± C/S) is compatible with very
low- to low-grade conditions. The K-white mica "crystallinity" values (0.20-0.26 Δ°2θ;
average 0.23) point to lower epizone conditions, very close to the boundary with the



anchizone (≈300 °C; Frey, 1987; Kisch, 1987). The temperatures calculated by RSCM in two
samples (315 and 330 °C) are compatible with the KI data of XRD analysis.
Our metamorphic data corroborate the existence of an unconformity between the lower and
upper formations of the Pulo do Lobo belt (Pérez-Cáceres et al., 2015). The lower formations
record a Devonian tectonothermal event that reached epizone or lower greenschist facies
conditions ($M_1$ with generalized phyllosilicate growth at temperatures as high as 450 °C),
while the overlying upper formation records a middle/upper Carboniferous tectonothermal
event close to the anchizone/epizone boundary ($M_2$ with small-sized phyllosilicate growth at
temperatures ≈300-330 °C; Table 1). Obviously, $M_2$ also affected somehow the lower
formations, being, at least in part, the responsible for the observed retrogression of $M_1$
chlorite and/or crystallization of new chlorites at lower temperature.

**5.4. Pressure conditions**
The measured $b$-cell parameters of K-white mica (in a short range between 8.991-9.002 Å;
average 8.996; standard deviation 0.003) are very similar in the lower and upper formations
of the Pulo do Lobo belt. Thus, the $b$ parameter is consistently homogeneous and reflects
very low phengite substitution in mica, as expected at low-pressure settings (Potel et al., 2006,
2016), near the intermediate pressure gradient boundary (Guidotti and Sassi, 1986).
In agreement with the low $b$-cell parameters, the composition of K-white mica is close to
muscovite with very low celadonite and higher pyrophyllite content (Fig. 5a), as expected for
illite-rich mica formed at low-pressure gradients. In the case of high- or medium-pressure
conditions, a continuous trend in mica compositions would be found reflecting the
decompression path after the peak pressure, while the $b$-cell parameter would represent an
average value of the range of mica compositions found in the sample (Abad et al., 2003b).
On the contrary, at low-pressure settings, the overall range of recorded pressure is very short
and micas present similar compositions and $b$-cell parameters among the various samples, as
in the case of the Pulo do Lobo samples (Figs. 5a and 6, and Table 1).
The Pulo do Lobo belt has been classically interpreted as a pre-collisional subduction-related
accretionary prism, based on the MORB geochemistry of their mafic rocks (see section 2.1).
According to this classical interpretation, features typical of modern subduction systems
should be expected, such as high-pressure metamorphic gradient remnants of partial
subduction/exhumation in an accretionary wedge (e.g., Platt, 1986; Ernst, 2005), or slices of
oceanic slab-derived lithologies (varied mid-ocean ridge metaigneous lithologies and also
deep ocean bottom metasediments). Thus, recent works on the Makran accretionary prism
(Omrani et al., 2017) and the subduction system of Japan (Endo and Wallis, 2017) describe
an accretionary mélange complex composed of pelagic sedimentary rocks, ophiolites,
greenschists, amphibolites, and blueschists with high-pressure minerals such as lawsonite and
glaucophane. On the contrary, most of the geological data concerning the Pulo do Lobo belt
do not back up such interpretation (see section 2.1), and our new results about pressure
conditions are also in disagreement. The only suspect of high-pressure gradient in the Pulo
do Lobo belt is the interpretation of some rhomboidal aggregates of epidote porfiroblasts as
the remnants of supposed lawsonite grown previously to $S_2$ in some samples of Pulo do





Lobo mafic schists (Rubio Pascual et al., 2013). However, no analytical data have been
presented to support the lawsonite pseudomorphs.

## 6. Conclusions

Eighteen samples of metapelites from the Pulo do Lobo belt have been studied to
characterize their Variscan low-grade metamorphism. The microstructural analysis of the
samples of the lower formations (Devonian Pulo do Lobo and Ribeira de Limas) shows the
existence of two superposed low-grade tectonothermal events with associated foliation and
phyllosilicate growth ($S_1$-$M_1$ and $S_2$-$M_2$; Table 2). $M_2$ was less intense, being the only event
that affected the overlying Carboniferous Santa Iría formation. The regional geology also
shows that a Mississippian thermal (magmatic-derived) event occurred in-between $M_1$ and
$M_2$.
$M_1$ and $M_2$ correspond to the chlorite zone, but $M_1$ entered the epizone (greenschists facies
with temperatures up to ≈450 °C), while $M_2$ did not exceed the anchizone-epizone boundary
(≈300 °C).
The temperatures obtained from RSCM are higher compared to the ones derived from
chlorite geothermometry and white mica data. The discrepancy can be explained by the fact
that RSCM records the true maximum temperature, being not affected by retrogression as
other methods do. In addition, this difference can be the consequence of the high sensitivity
of CM to quickly equilibrate at maximum temperatures during short thermal events due to
magmatic intrusions emplaced during the Mississippian thermal event.
Thermodynamic disequilibrium between white mica and chlorite has precluded their use for
geothermobarometry, and a variety of data (including the existence of relic high-temperature
chlorite cores, the presence of chlorite/smectite mixed layers, or the very-low temperatures
calculated with chlorite geothermometers) indicate chlorite retrogression after $M_1$
metamorphic climax and crystallization of new chlorite grains at lower temperature.
The low-pressure conditions derived from white mica indicators (very low celadonite content
and $b$-cell values) are incompatible with the high-pressure metamorphic gradient expected in
a subduction-related accretionary wedge, which has been the classical interpretation of the
Pulo do Lobo belt.


## Acknowledgements

This work was supported by the projects CGL2011-24101 (Spanish Ministry of Science and
Innovation), CGL2015-71692-P and CGL2016-75679-P (Spanish Ministry of Economy and
Competitiveness), RNM-148 and RNM-179 (Andalusian Government) and BES-2012-
055754 (Doctoral scholarship to I. Pérez-Cáceres from the Spanish Ministry of Science and
Innovation). The Raman facility in Paris has been funded by the City of Paris (Emergence
program). We thank Valérie Magnin for her assistance with the microprobe analysis in
Grenoble and Pierre Lanari for his support with thermodynamic software.





**Figure captions**

**Figure 1**. a) Location of the studied area in the SW of the Iberian Massif (in grey). CIZ: Central Iberian Zone, OMZ: Ossa-Morena Zone, SPZ: South Portuguese Zone. b) Geological map of the Pulo do Lobo belt and other units related to the OMZ/SPZ boundary with indications of the two cross-sections studied. c.1-2) Geological cross-sections of the Pulo do Lobo belt (see b for location) (modified from Martínez Poza et al., 2012 and Pérez-Cáceres et al., 2015). Numbered red circles in b-c locate the samples studied. Big circles show the KI values for 10 Å reflection peaks of K-white mica and the average RSCM temperatures, with the relative colour bar according to the results shown in Table 1. BAA: Beja-Acebuches Amphibolites, M: metabasalts, PL: Pulo do Lobo formation, RL: Ribeira de Limas formation, SI: Santa Iría formation.

**Figure 2**. Pictures of the Pulo do Lobo rocks illustrating deformation at outcrop scale: a) Pulo do Lobo formation, b) Ribeira de Limas formation, c) Santa Iría formation. Microphotographs from thin-sections: d) Cross-polarized light image of sample PLB-84 (Pulo do Lobo formation), e) SEM-BSE image of sample PLB-88 (Ribeira de Limas formation), f) Cross-polarized light images of sample PLB-71 (Santa Iría formation).

**Figure 3**. Representative Raman spectra of CM across the Pulo do Lobo belt from low temperature (bottom; Santa Iría formation) to high temperature (top; lower formations) including the average maximum temperatures (ºC) for each sample. Vertical scale for spectrum intensity is arbitrary. See Fig. 1 for sample location and Table 1 and supplementary information for RSCM data.

**Figure 4**. X-Ray maps of the three selected samples analyzed by EPMA and processed with XMapTools. The samples belong to the lower formations of the Pulo do Lobo belt (sample PLB-88: Ribeira de Limas formation; samples PLB-84 and PLB-93: Pulo do Lobo formation; the latter (PLB-93) is close to Early Carboniferous igneous intrusions). a) EPMA BSE photographs. b) Mineral maps. c) $Al^{IV}$ content map in chlorites, which increases with temperature. The white square highlights the zonation of a chlorite grain from core to rim. d) Temperature maps of chlorite using the Lanari et al. (2014a) geothermometer assuming all iron as ferrous. White squares show selected areas illustrating higher-temperature chlorite cores. Red squares show the selected areas (representative of $S_1$ foliation) used for chlorite-quartz-water geothermometric calculations shown in Fig. 7.

**Figure 5**. Ternary plots of all the analyzed white micas (a) (Cel: celadonite, Mus: muscovite, Prl: pyrophyllite) and chlorite (b) (Cli+Daph: clinochlore + daphnite, Am: amesite, Sud: sudoite) plotted with the XmapTools TriPlot3D module. Colour bars refer to the number of mica/chlorite pixels analyzed.

**Figure 6**. Compositional diagram of white micas showing Na/Na+K vs Si/Al (atomic ratios) for 31 EPMA point analyses from seven samples of the lower formations of the Pulo do Lobo belt (different symbology, for each sample). Point analyses were obtained on the microprobe at the University of Huelva (Spain). Qualitative information about temperature and pressure conditions are respectively according to Guidotti et al. (1994), Coggon and Holland (2002), Parra et al. (2002), Massonne and Schereyer (1987) and Massonne and Szpurka (1997).

**Figure 7**. Histograms of temperatures obtained using the chlorite-quartz-water geothermometer of Vidal et al. (2006) (a) and Bourdelle et al. (2013) (b) on selected representative $S_1$ chlorites (see red squares in Fig. 4d for location). *n* represents the number of chlorites that could be used for each calibration. The number of analyses is lower in those with Vidal et al. (2006) approach because the assumption that the Si content of chlorite is lower than 3 apfu.



**Table captions**
**Table 1**. Samples and results obtained by XRD (<2 μm fraction), white mica and chlorite
compositions, temperature ranges from chlorite thermometry, and average RSCM thermometry. KI
values and average RSCM temperatures show a relative colour-bar scale. Mineral abbreviations
according to Whitney & Evans (2010). Qz: Quartz, Ms: Muscovite, Fsp: Feldespar, Chl: Chlorite, Pg:
paragonite, C/S: chlorite-smectite mixed layers, Cel: celadonite, Prl: pyrophyllite, Cli+Daph:
clinochlore + daphnite, Am: amesite, Sud: sudoite, Std Dv: standard deviation.
**Table 2.** Summary of the tectonometamorphic Variscan evolution of the Pulo do Lobo belt.

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



Figure 1



Figure 2



Figure 3

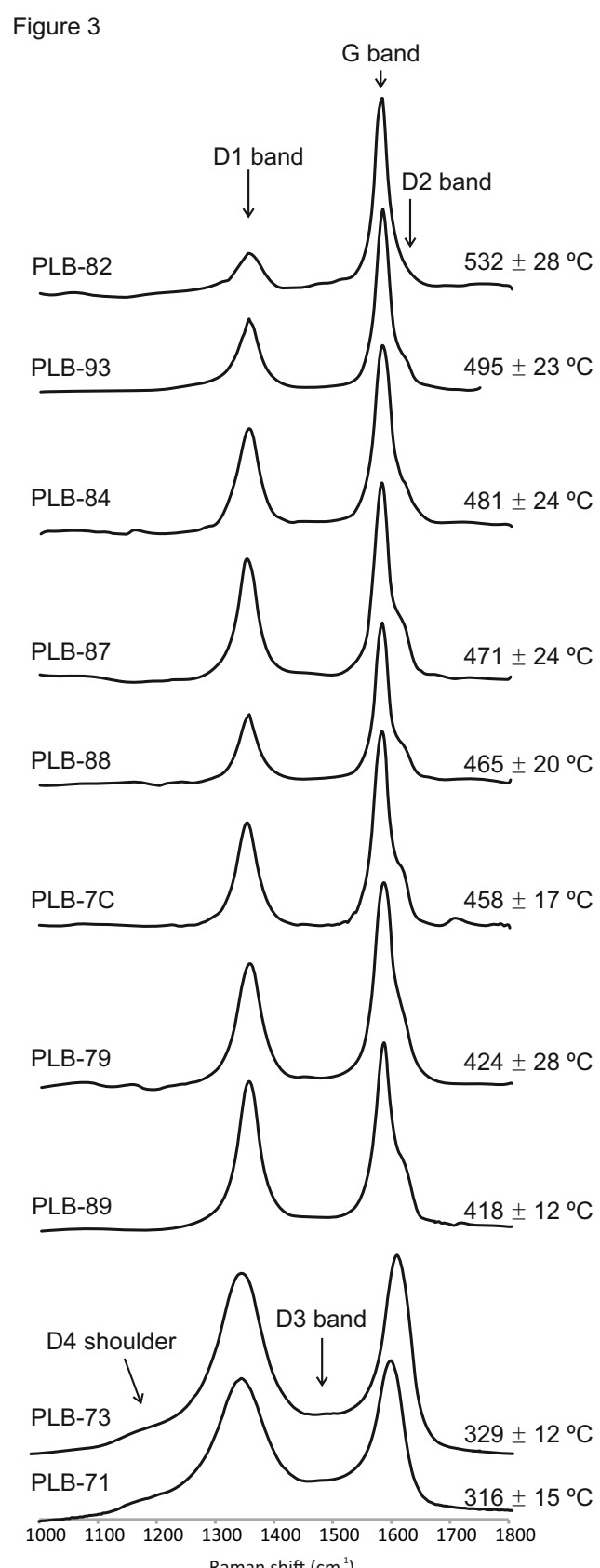



Figure 4





Figure 5

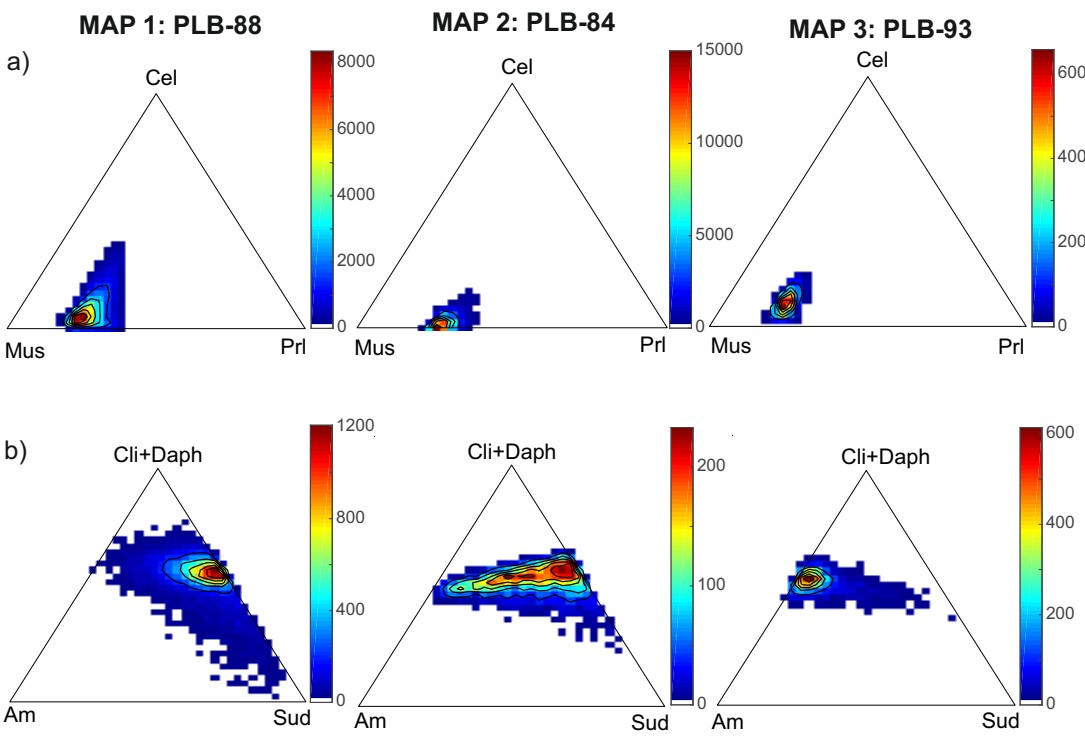





Figure 6

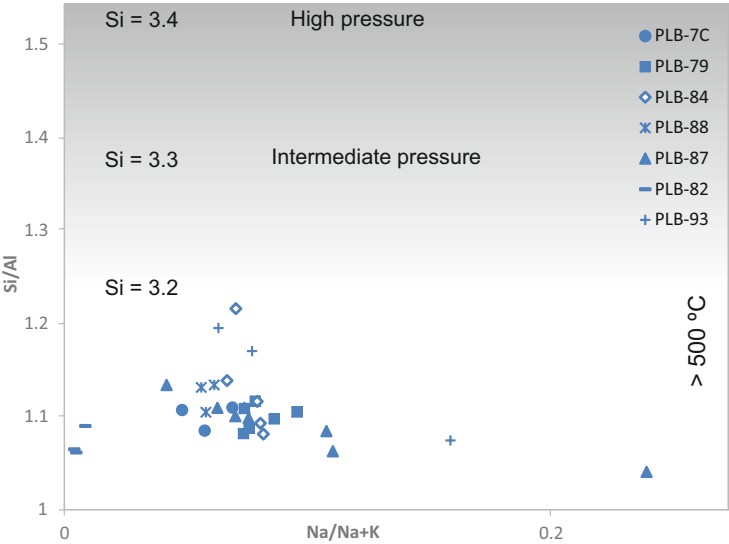





Figure 7

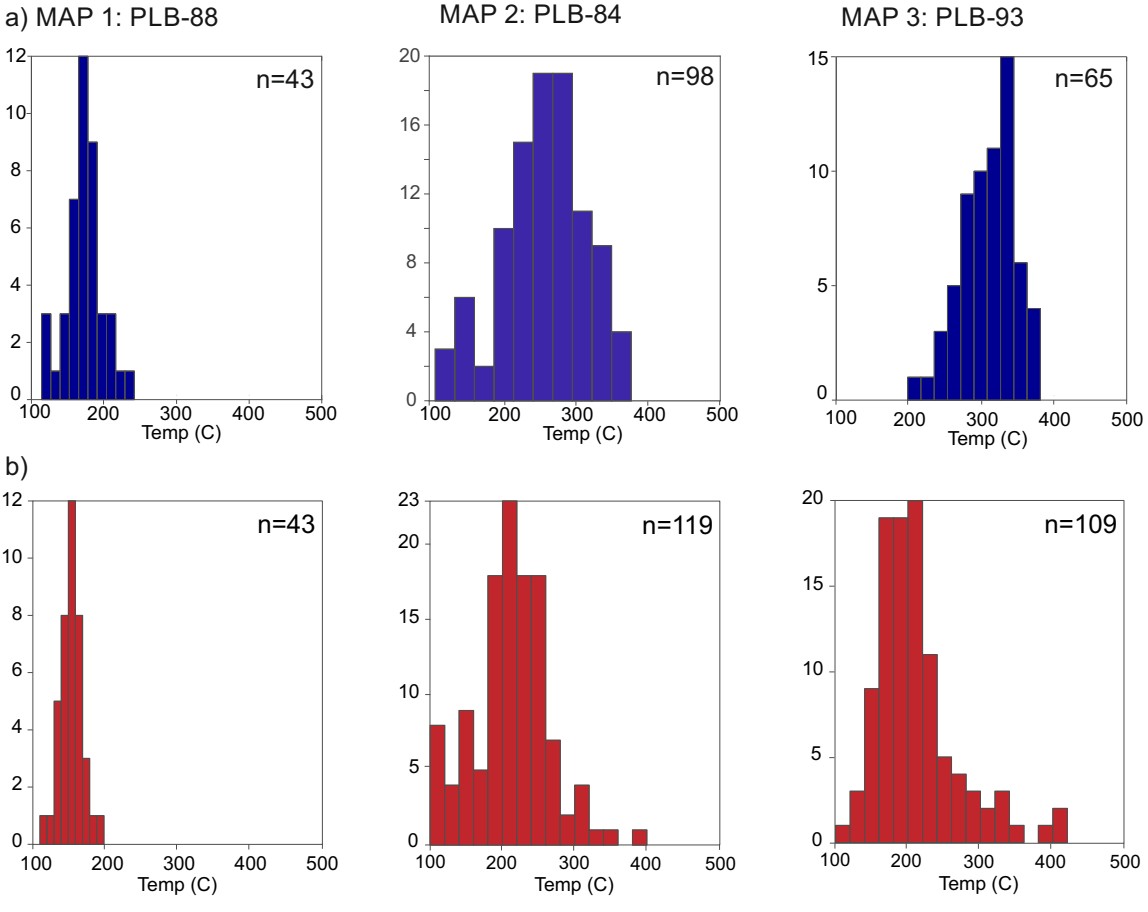





Solid Earth Discussions — Open Access — EGU

**Table 1**

| Formation | Sample PLB- | Mineralogy Qz + Ms + Fsp+ | FWHM A | Basel KI (10 A) bulk fraction | Basel KI (10 A) <2 μm | b (A) Ms | d001 (A) Ms | White mica compositions % Ms | % Cel | % Prl | Chlorite compositions % Cli+Daph | % Am | % Sud | Chlorite maps (Lanari et al., 2014b) T (°C) | Chlorite thermometry Vidal et al., 2006 T (°C) | Bourdelle et al., 2013 T (°C) | Tmax (°C) RSCM Mean | Std Dv |
|---|---|---|---|---|---|---|---|---|---|---|---|---|---|---|---|---|---|---|
| Santa Iría (upper formation) | 71 | Chl | 0.221 | 0.23 | 0.25 | 8.991 | 9.995 | | | | | | | | | | 316 | 15 |
| | 73 | Chl | 0.227 | 0.22 | 0.26 | 8.996 | 9.997 | | | | | | | | | | 329 | 12 |
| | 74 | Chl + C/S | 0.164 | 0.20 | 0.20 | 8.999 | 10.001 | | | | | | | | | | - | |
| | 76 | Chl | 0.184 | 0.20 | 0.22 | 8.997 | 9.997 | | | | | | | | | | - | |
| | 77 | Chl + C/S | 0.171 | 0.19 | 0.21 | 8.998 | 9.995 | | | | | | | | | | - | |
| | 79 | Chl + Pg | 0.17 | 0.18 | 0.20 | 8.995 | 9.993 | | | | | | | | | | 424 | 28 |
| | 80 | Chl | 0.169 | 0.18 | 0.20 | 9.001 | 9.988 | | | | | | | | | | - | |
| | 81 | Chl + Pg + C/S | 0.181 | 0.19 | 0.22 | - | 9.988 | | | | | | | | | | - | |
| | 82 | Chl + Pg | 0.158 | 0.17 | 0.19 | 8.995 | 9.986 | | | | | | | | | | 532 | 28 |
| lower fomations | 84 (map 2) | Chl + Pg + C/S | 0.173 | 0.17 | 0.21 | 8.994 | 9.988 | 70-80 | 0-10 | 20-30 | 50 | 0-50 | 0-50 | 150-350 | 150-375 | 150-350 | 481 | 24 |
| | 85 | Chl + Pg | 0.137 | 0.17 | 0.18 | 8.996 | 9.996 | | | | | | | | | | - | |
| | 86 | Pg + C/S | 0.144 | 0.18 | 0.19 | 8.993 | 9.986 | | | | | | | | | | - | |
| | 87 | Chl + Pg | 0.144 | 0.18 | 0.19 | 8.998 | 9.986 | | | | | | | | | | 471 | 24 |
| | 88 (map 1) | Chl + Pg | 0.129 | 0.18 | 0.17 | 8.997 | 9.99 | 70-80 | 0-10 | 20-30 | 50 | 0-10 | 20-50 | 100-200 | 120-230 | 150-200 | 465 | 20 |
| | 89 | Chl | 0.178 | 0.19 | 0.21 | 8.996 | 9.993 | | | | | | | | | | 418 | 12 |
| | 91 | Chl + Pg | 0.143 | 0.17 | 0.19 | 9 | 9.995 | | | | | | | | | | - | |
| | 93 (map 3) | Chl + Pg | 0.128 | 0.18 | 0.17 | 9.002 | 9.99 | 70-80 | 0-10 | 20-30 | 50 | 40-50 | 0-10 | 200-450 | 200-380 | 150-400 | 495 | 23 |
| | 7C | - | - | - | - | 8.993 | - | | | | | | | | | | 458 | 17 |



**Table 2**

| Time | Deformation/metamorphic phase | Temperature | Low-grade metamorphic conditions |
|---|---|---|---|
| Middle-Upper Carboniferous | $S_3$ <br> $S_2$-$M_2$ | - <br> <300 °C | - <br> Epizone-Anchizone limit |
| Early Carboniferous (~340 Ma) | Beja-Acebuches and Pulo do Lobo metamafics <br> Thermal imprint | | |
| Upper Devonian | $S_1$-$M_1$ | ~300-450 °C | Epizone |