# Peer review of "Deciphering the metamorphic evolution of the Pulo do"

_Solid Earth, 2019_

## Referee Comment (RC1) · Cecilio Quesada (Referee) · 3 Nov 2019

1. General comments This paper constitutes a very well-documented study of the characteristics of phyllosilicates and organic matter in pelitic samples collected along two sections across the Pulo do Lobo "belt", a critical unit to understand the so-called SW Iberia Variscan suture. The goal and approach are to be applauded and the paper deserves publication. That said, I find the results to be incomplete, mainly because the short number of samples in just two sections located near the eastern end of a belt that is much larger, wider and better exposed in its central and western parts in Portugal. The conclusions are supported by the data presented but I would like to see results

from other sections and from other rock types, for instance the metamafic rocks that are only marginally mentioned. One of the conclusions reached by the authors is that there is in no evidence for HP metamorphic conditions in their samples. Fair enough! But surprisingly they have not attempted to analyze samples in the vicinity of the only reference to possible HP relics published so far (Rubio Pascual et al., 2013). I recognize the interest of the data presented and recommend publication but at the same time I insist that the authors should enlarge the scope of their research as suggested above. Also with a general character, I find the paper to present only the authors' interpretation of the SW Iberia Variscan orogen. This is evidenced by many references to the first author's papers and those of her group, hiding other controversial (to the authors' minds) interpretations behind the general statement "and references therein". This would be fine if they did not derived conclusions that have profound geodynamic implications. Consensus is far from being reached on the geodynamic evolution of this part of the Variscan orogen, and the authors' model is just one of many. In this respect, the entire "Section 2: Geological setting" is rather disappointing. Potential readers would welcome a discussion of the various models proposed, at least on those aspects that are later discussed in the paper.

2. Specific comments References cited in the comments below and not included in the paper are listed at the end of this section. A) Lines 150-152: The authors should follow consistent criteria to describe the subdivision of the South Portuguese Zone (SPZ). As written, they are mixing structural and stratigraphic criteria (Pulo do Lobo and Iberian Pyrire belts are structural units whereas the Carboniferous flysch refers to a stratigraphic unit). My recommendation is to use a first order structural division of the entire SPZ, not only of the units exposed in Spain, and eventually a reference to the stratigraphy of each of them. Flysch units occur in all structural divisions and those present in the Pulo do Lobo belt are not Carboniferous but Late Devonian (Famennian) in age, at least in part. B) Section 2.1. Pulo do Lobo Belt. The authors refer in this section to only a part of the stratigraphic units described in the Pulo do Lobo Belt in Spain. In my opinion they should refer to all the stratigraphic divisions that crop out

in both Spain and Portugal. By the way, I am respecting in this review the term "belt", as a simply descriptive word, despite this unit has been referred to with other more genetic terms (zone, terrane). A critical review of this topic would be appreciated by potential readers, mainly if the Pulo do Lobo belt is thought to occur in the vicinity of a major suture zone between Laurussia (Avalonia) and Gondwana, as inferred in section 2. C) Lines 181-191: The paleontologically-based Famennian age of the Santa Iria formation is disputed on the basis of younger detrital zircon ages. See below the discussion in point (v) of comment D. Most authors would agree that the Santa Iria formation was only affected by the 3rd regional foliation (a pressure/solution type). See Braid et al., 2010 and Quesada et al., 2019 for description and further references. D) Lines 197-213: Biased and incomplete description of the mafic rocks in the Pulo do Lobo belt and their possible interpretation. The authors are referred to the chapter by Quesada et al., 2019 for a more complete description. The following points deserve special attention: i) On lines 199-200 the authors write "interpreted as a tectonic mélange (the so-called Peramora Mélange; Fig. 1b-c; Apalategui et al., 1983; Eden, 1991; Dahn et al., 2014)". The term Peramora mélange was first introduced by Eden, 1991 in his thesis; therefore the reference to Apalategui et al., 1983 is inappropriate here. ii) In addition to the tectonic nature of the imbrication of the mafic rocks with Pulo do Lobo schists as well as internally, those authors also recognized the sedimentary nature of the Peramora mélange, which consists of both mafic matrix and mafic and sedimentary clasts (from mm-to m-scale, lenticular to sigmoidal-equidimensional in shape). Mafic clasts include basalts, dolerites and gabbros, some of the latter preserving the original igneous textures and no pervasive cleavage, although all the rocks in the mélange are retrogressed to greenschist facies conditions. iii) MORB-like metamafic rocks also occur in the Portuguese extension of the Pulo do Lobo belt, namely along the Trindade-Alfarrobeira strip, which are intersected by the Alfarrobeira drill core (see Ferreira and Oliveira in Quesada et al., 2019). There, a more coherent metamafic package is imbricated with minor Pulo do Lobo schist as well as internally. Individual metamafic horses reach up to decametric thicknesses (the interpretation as a tectonic

mélange seems plausible here!). iv) The rocks referred to above are not the only igneous rocks present in the Pulo do Lobo belt. Especially relevant to this discussion paper is, the ca. 354 Ma calcalkaline, subduction-related gabbro component of the Gil Márquez pluton (Gladney et al., 2014; Braid et al., 2018). These rocks are unfoliated and intruded Pulo do Lobo belt rocks (but not the Santa Iria formation) after the first two phases of cleavage development recorded in these rocks (presumably S1 and S2 in the discussed paper). The arc signature of the gabbros indicates that subduction was still active in this part of the orogen at ca. 354 and that the Pulo do Lobo belt was located in the upper plate. These data supports the interpretation of the belt as part of an accretionary prism. v) The youngest U-Pb zircon ages (mostly LA-ICPMS data)obtained from the metamafic igneous rocks are taken as representing maximum crystallization ages, implying that their emplacement must postdate deposition of the oldest metasedimentary rocks in the belt, therefore, they must be intrusive. Several problems arise, though. First, these young ages occur in both matrix and clasts of the mafic mélange as well as in the metasediments, the palynomorph-derived Devonian age of which is not disputed by the authors (the only exception being the Santa Iria formation, whose age is also disputed). Second, the metamafic rocks (at least the matrix) are also deformed by the same three folding events recorded in the metasediments. Third, intrusion of the ca. 354 Ma (also an LA-ICPMS age) unfoliated gabbro of the Gil Márquez pluton postdates the first two cleavages in the host rocks. All these data make it likely that a partial rejuvenation of the U-Pb system in zircons may have taken place, as already interpreted by Dahn et al., 2014. This rejuvenation may have been related to: i) subsequent syn-collisional slab breakoff (ca. 345-335 Ma) events, interpreted by Braid et al., 2018 on the basis of the ages and geochemical signatures of the main felsic components of the Gil Márquez pluton, which also intrude the Santa Iria Flysch, ii) a lithospheric delamination and mantle replacement event (ca. 316 Ma), interpreted by Dupuis et al., 2014 to account for the emplacement across SW Iberia of a swarm of mafic dikes with MORB signatures; and iii) a combination of those two or other processes not recognized as yet. vi) Mississippian magmatic rocks occur in profusion

across SW Iberia (from the Iberian Pyrite Belt in the SPZ, through the Pulo do Lobo belt, the OMZ and also the southern part of the CIZ). They occur however in rather different paleogeographic and paleotectonic environments that probably did not occupy the present-day relative positions in the Mississippian, mainly if the otherwise unknown but surely significant displacements during the left-lateral subduction/collision orogenesis are taken into account. Assuming that all these igneous rocks developed in response to a single event seems rather speculative. vii) The authors refer to "Peramora Olistostrome, Pérez-Cáceres et al., 2015", which is in fragant contradiction with their previous interpretation of the Peramora málange as a tectonic one (line 199). E) Line 219: Please specify formation names F) Lines 240-241: assuming that detrital micas are "generally larger than 2 $\mu$m" seems an oversimplification that might lead to misinterpretations. At smaller grain sizes it is not a simple task distinguishing syn-kinematic neoblasts from mechanically rotated detrital grains that may preserve a record of previous events under the prevailing very low- to low-grade metamorphic conditions. G) Line 467: The age attribution of S1 to the Devonian and S2 to the Carboniferous is arbitrary. Both cleavages formed prior to intrusion of the ca. 354 Ma gabbro of the Gil Márquez pluton (see Gladney et al., 2014; Braid et al., 2018). In addition, Braid et al., 2010 demonstrated that both cleavages and associated folds developed under a similar strain regime dominated by sinistral transpression, and suggested that they formed during a process of progressive deformation culminating in exhumation. Thus, S1 may have formed at certain depths during progressive burial, whereas S2 would have developed during subsequent exhumation. This interpretation would be in agreement with the metamorphic evolution described in this discussion paper. The problem in the authors' age attribution may reside is their interpretation of the youngest detrital zircons in the Santa Iria formation as maximum depositional ages, a point discussed by Dahn et al., 2014 and Oliveira et al., in Quesada et al., 2019, who interpret those younger ages to be the result of a process of partial lead loss (see point (v) in comment D). A discussion on this topic would be appreciated. H) Lines 487-489: There is ample consensus that the Santa iria formation is only affected by the 3rd deformation event,

with similar geometrical and kinematic characteristics throughout the area and different to those of D1 sand D2 (see Braid et al., 2010 and Quesada et al., 2019 for description and further references) . I) Lines 516-519: Samples PLB 93 (and aso PLB 91) appears to have been collected (lousy precision in Fig. 1) near to the contact with the Peramora mélange or within one of the metasedimentary horses imbricated with it. There, apart from the mafic sedimentary mélange (Peramora mélange s. str. or Peramora olistostrome sensu Pérez-Cáceres et al., 2015) and Pulo do Lobo schist imbricates there exists a swarm of late mafic dikes (those with MORB signature dated at ca. 316 Ma by Dupuis et al., 2014). These are beautifully exposed along the Alcalaboza river where they intrude the Peramora málange and imbricated Pulo do Lobo schist, and hand specimens of these basaltic rocks are hard to distinguish from unfoliated metamafic rocks of the clasts of the mélange. If the increase in T described by the authors was related to the emplacement of the dikes, those samples should not be considered further. J) Lines 542-544: Biotite was reported by Apalategui et al., 1983 and Braid et al., 2010, among others, in rocks belonging to the Pulo do Lobo and Ribeira de Limas formations. Reference to those works should be given, and discussed if the authors believe that biotite is not paragenetic with the other phyllosilicates. If biotite is a part of the syn-kinematic paragenesis, which I think is, then the following discussion in this paragraph should be reconsidered. K) Line 566 and section 5.3: The attribution to the "middle/upper Carboniferous" of this event is not supported by undisputable data. In addition, I am confused on the relationship between deformation and metamorphic events (D1-D2-D3 and M1-M2, in the authors' terminology). In the lack of reliable thermochronologic data I would like to see a clear statement on this issue. If as the authors state, and everybody agree!, the Santa Iria formation is unconformable on the previously deformed lower sequences, then the latter must have been exhumed prior to deposition of the former. Therefore, every metamorphic evidence supporting temperatures >300°C must relate to the burial of the Pulo do Lobo and Ribeira de Limas formations, that is, pre-late Famennian. L) Lines 609-613. I wonder why the authors have not tried to collect and analyze samples of metasedimentary rocks in the area

where those pseudomorphs were reported. It would have been worthwhile! M) Lines 616-623: Where has S3/D3 (lines 178 and 191) gone? What is its role in the story? N) Lines 638-641: The lack (or poor preservation) of HP characteristics may have an explanation in various well-documented processes: i) the predominant sinistral strike-slip regime of the deformation in this part of the orogen (large lateral displacements vs. little burial); ii) thermal overprint and re-equilibration after accretion to the upper plate as suggested by intrusion of the calkalkaline arc-related gabbros at ca. 354 Ma; iii) thermal overprint and re-equilibration during subsequent emplacement of late- to post-kinematic igneous rocks (e.g. Sierra Norte Batholith); etc.

References cited in the comments but not in the paper - Braid JA, Murphy JB, Quesada C, Gladney ER, Dupuis N (2018) Progressive magmatism and evolution of the Variscan suture in southern Iberia. Int J Earth Sci (Geol Runds) 107: 971–983. - Dupuis NE, Braid JA, Murphy JB, Quesada C, McFarlane CRM (2014) Changing mantle sources in a suture zone in the heart of Pangea: implications for collisional tectonics during the waning stages of ocean closure. Int J Earth Sci (Geol Runds) 103: 1403-1414. - Gladney ER, Braid JA, Murphy JB, Quesada C, McFarlane CRM (2014) U-Pb geochronology and petrology of the late Paleozoic Gil Márquez pluton: Magmatism in the Variscan suture zone, southern Iberia, during continental collision and the amalgamation of Pangea. Int J Earth Sci (Geol Runds) 103:1433–1451. Quesada C., Braid JA, Fernandes P, Ferreira P, Jorge RS, Matos JX, Murphy JB, Oliveira JT, Pedro J, Pereira, Z (2019) SW Iberia Variscan Suture Zone: Oceanic Affinity Units. In: Quesada C and Oliveira JT (Eds), The Geology of Iberia: A Geodynamic Approach, v. 2: 131-171. Springer Regional Geology Reviews.

3. Technical corrections Lines 99-100: Insert "to" between "allows" and "know" Line 129: Write "At present" instead of "Actually" Line 134: Write "that" between "shearing" and "occurred" Line 221: write "non-altered" instead of "not altered" Line 539: write "grains" instead of gains"

---

## Referee Comment (RC2) · Anonymous Referee #2 · 14 Nov 2019

1. General comments. This paper focuses on the characterization of the low-grade metamorphic conditions of the Pulo do Lobo Zone. For that, 18 metapelites were studied with several techniques: X ray diffraction (mineralogy, KI and b), RSCM, EMPA and chlorite geothermometry. The results of this study have allowed to correlate two phyllosilicate growth events with different deformation/metamorphic phase and to establish the PT conditions of these events with more detail than the more general previous works related to the South Portuguese Zone. The results are shown clearly, what facilitates the reading and understanding of the article (discussion and conclusions). In any case, there are some aspects that can be improved. Some suggestions and little corrections are indicated below and some others are included in a pdf file.

[Figure]

2. Specific comments. At the end of the abstract, lines 48-49: please, include data: the range or an indication of the celadonite content (e.g. <X apfu), the range or average of b dimension and something more specific for the low P gradient. Keywords: Include a keyword relative to the illite "crystallinity". As for section 3 (Samples and analytical methods): Authors should add some information in Methods section about the Scanning Electron Microscope used. The information relative to the EMPA at the University of Huelva is too short in comparison with the EMPA study in Grenoble. Please, add information about analyses time and standards. Line 276: Which T of formation? the T of mica formation? Lines 341-344: if you say this, please justify or give the reason why two of them were excluded. As for section 4 (Results) Lines 369-370: Is this observation important for something? It is not very coherent with the comment in lines 241-242. In relation to the comment in lines 374-376, please add a column in Table 1 with the KI corresponding to the bulk fraction. Lines 393-395: As for the KI you have low-grade metamorphic conditions (epizone). And C/S is compatible also with low grade metamorphic conditions as you say also in line 493. So, I recommend to delete "very low-" and even "the presence of C/S". Line 395: in general, you use frequently "low-P metamorphic gradient" along the paper, please be more specific, what do you want to say with low-pressure gradient? Line 412: Please indicate with a value (< X Å) what it means low b parameter and the same for the d001 (> X Å). Lines 419-420: Could you justify this assertion better? Only because is poor in sudoite? Poor in sudoite but higher in... Lines 443-447: Please check the T ranges and several comments included in the pdf file revised. Discussion Line 471: Table 2 should be cited later, at the end of 5.3 section for example. In lines 490-491 I think the comment it is not significant. As well, as far as I know the Bourdelle thermometer is mainly calculated for low P and low T (< 350°C) so these results should not be considered. Line 498: this value (KI 0.14) is the boundary of something? I feel lost. Clarify in the text. Lines 511-513: These temperature ranges based on several methods (figs 4d and 7 and Table 1) are not very clear. How have you selected these final ranges from three intervals (one for each method)? In sample 84 you have selected from the lowest T to the highest one, but in the two other samples? Lines 530-532: This parameter (KI) is used for low-grade metamorphism, you are talking about T much higher (until 450°C!!) than the estimations we can do with the illite crystallinity (epizone, 300-350 °C) The use of colors in the table 1, although it said it is a relative colour bar, is a bit confusing considering that KI and RSCM give different T ranges for the lower formations. Figure 7: Why are not used the same ranges of temperatures for red and blue hystograms? The blue columns are wider..so the comparison is not straight.

3. Technical corrections. Line 99-100: "allows to know" Lines 292-293: Please use lowercase as in line 300 Line 361: According

Please also note the supplement to this comment:
https://www.solid-earth-discuss.net/se-2019-143/se-2019-143-RC2-supplement.pdf

**Supplement:**

[revised manuscript text omitted]

Figure 1

[Figure]

[Figure]

Figure 2

[Figure]

Figure 3

[Figure]

[Figure]

[Figure]

Figure 4

MAP 1: PLB-88    MAP 2: PLB-84    MAP 3: PLB-93

[Figure]

Figure 5

[Figure]

[Figure]

Figure 6

[Figure]

[Figure]

[Figure]

Figure 7

[Figure]

[Figure]

[Figure]

Solid Earth Open Access
Discussions
EGU

**Table 1**

[Figure]

| Formation | Sample PLB- | Mineralogy Qz + Ms + Fsp+ | FWHM Å | Basel KI (10 Å) bulk fraction | Basel KI (10 Å) <2 µm | b (Å) Ms | d001 (Å) Ms | White mica % Ms | White mica % Cel | White mica % Prl | Chlorite % Cli+Daph | Chlorite % Am | Chlorite % Sud | Chlorite maps (Lanari et al., 2014b) T (°C) | Vidal et al., 2006 T (°C) | Bourdelle et al., 2013 T (°C) | Tmax RSCM Mean | Tmax RSCM Std Dv |
|---|---|---|---|---|---|---|---|---|---|---|---|---|---|---|---|---|---|---|
| Santa Iría (upper formation) | 71 | Chl | 0.221 | 0.23 | 0.25 | 8.991 | 9.995 | | | | | | | | | | 316 | 15 |
| | 73 | Chl | 0.227 | 0.22 | 0.26 | 8.996 | 9.997 | | | | | | | | | | 329 | 12 |
| | 74 | Chl + C/S | 0.164 | 0.20 | 0.20 | 8.999 | 10.001 | | | | | | | | | | - | |
| | 76 | Chl | 0.184 | 0.20 | 0.22 | 8.997 | 9.997 | | | | | | | | | | - | |
| | 77 | Chl + C/S | 0.171 | 0.19 | 0.21 | 8.998 | 9.995 | | | | | | | | | | - | |
| | 79 | Chl + Pg | 0.17 | 0.18 | 0.20 | 8.995 | 9.993 | | | | | | | | | | - | |
| | 80 | Chl | 0.169 | 0.18 | 0.20 | 9.001 | 9.988 | | | | | | | | | | 424 | 28 |
| | 81 | Chl + Pg + C/S | 0.181 | 0.19 | 0.22 | - | 9.988 | | | | | | | | | | - | |
| | 82 | Chl + Pg | 0.158 | 0.17 | 0.19 | 8.995 | 9.986 | 70-80 | 0-10 | 20-30 | 50 | 0-50 | 0-50 | | | | 532 | 28 |
| | 84 (map 2) | Chl + Pg + C/S | 0.173 | 0.17 | 0.21 | 8.994 | 9.988 | 70-80 | 0-10 | 20-30 | 50 | 0-50 | 0-50 | 150-350 | 150-375 | 150-350 | 481 | 24 |
| | 85 | Chl + Pg | 0.137 | 0.17 | 0.18 | 8.996 | 9.996 | | | | | | | | | | - | |
| | 86 | Pg + C/S | 0.144 | 0.18 | 0.19 | 8.993 | 9.986 | | | | | | | | | | - | |
| lower formations | 87 | Chl + Pg | 0.144 | 0.18 | 0.19 | 8.998 | 9.986 | 70-80 | 0-10 | 20-30 | | | | | | | 471 | 24 |
| | 88 (map 1) | Chl + Pg | 0.129 | 0.18 | 0.17 | 8.997 | 9.99 | | | | 50 | 0-10 | 20-50 | 100-200 | 120-230 | 150-200 | 465 | 20 |
| | 89 | Chl | 0.178 | 0.19 | 0.21 | 8.996 | 9.993 | | | | | | | | | | 418 | 12 |
| | 91 | Chl + Pg | 0.143 | 0.17 | 0.19 | 9 | 9.995 | | | | | | | | | | - | |
| | 93 (map 3) | Chl + Pg | 0.128 | 0.18 | 0.17 | 9.002 | 9.99 | | | | 50 | 40-50 | 0-10 | 200-450 | 200-380 | 150-400 | 495 | 23 |
| | 7C | - | - | - | - | 8.993 | - | | | | | | | | | | 458 | 17 |

[Figure]

**Table 2**

| Time | Deformation/metamorphic phase | Temperature | Low-grade metamorphic conditions |
|---|---|---|---|
| Middle-Upper Carboniferous | $S_3$

$S_2$-$M_2$ | -

<300 °C | -

Epizone-Anchizone limit |
| Early Carboniferous (~340 Ma) | Beja-Acebuches and Pulo do Lobo metamafics

Thermal imprint | | |
| Upper Devonian | $S_1$-$M_1$ | ~300-450 °C | Epizone |

---

## Author Comment (AC1) · 16 Dec 2019

Reply to the interactive comment of referee#1 (Cecilio Quesada) on "Deciphering the metamorphic evolution of the Pulo do Lobo metasedimentary belt (SW Iberian Variscides)" by Irene Pérez-Cáceres et al. (Manuscript number se-2019-143).

We acknowledge the interactive comment made by Cecilio Quesada as referee. His suggestions have contributed to clarify some issues of the manuscript, which are now included in the revised version. This review is focused on disputable interpretations of the regional geology rather than on the main topic of our manuscript. All of these regional issues are answered in the following paragraphs, though we have reorganized

and grouped them in order to avoid unnecessary repetitions. The line numbers quoted correspond to the revised version of the manuscript with tracked changes (uploaded as supplementary file to this response).

1. Geological setting. A more inclusive geological setting has been made by adding sentences and new references to authors with interpretations complementary or alternative to ours (lines 110-111, 128-129, 140-142).

2. Division in units of the SPZ. Regarding nomenclature, we prefer to use the term "Pulo do Lobo" in a merely descriptive way. Thus, the term belt was used in the submitted version of our manuscript; nevertheless, and in order to avoid any confusion (as claimed by the reviewer), we have renamed now the two major units of the region as: "Pulo do Lobo domain" and "SPZ domain" (lines 160-162). The subdivision of these two major domains is as follows, according to geological mapping both in Spain and Portugal: a) The Pulo do Lobo domain includes, from bottom to top, the following stratigraphic formations: i) the Pulo do Lobo Fm; ii) the Ribeira de Limas Fm; and iii) the Santa Iría Fm, which unconformably overlies the other two formations. The relatively minor mafic rocks are embedded in the Pulo do Lobo Fm. b) The SPZ domain includes, from bottom to top: i) the Phyllite-Quartzite (PQ) Group; ii) the Volcano-Sedimentary Complex (CVS); and iii) the Culm or Flysch Group. The southern ZSP is dominated by the Flysch Group, except at the southernmost corner where underlying formations equivalent to the PQ and CVS crop out. We realize that the Santa Iría Fm (Pulo do Lobo domain) and the Flysch Group of the SPZ might be considered as a single tectonosedimentary unit with younger ages southwards. However, in order to comprehensively describe the tectonostratigraphic evolution of each domain (particularly the Pulo do Lobo domain; section 2.1 of the manuscript), the division shown above seems preferable.

3. Age of the Santa Iría Fm. There are two sources of evidence for the age of the Santa Iría Fm. On the one hand, palynological data (Pereira et al., 2018) suggest a Late Famennian age. On the other hand, detrital zircon populations (Braid et al., 2011;

Pérez-Cáceres et al., 2017) point to an early Carboniferous age. The latter age seems preferable due to the common palynomorph reworking shown in some papers (Lopes et al., 2014). Alternatively, partial rejuvenation of the zircon U/Pb system during low-grade metamorphism has been claimed to discard the sedimentary early Carboniferous age, though the detrital zircon populations are robust and highly concordant (particularly SHRIMP data). Anyway, the exact age of the Santa Iría Fm is neither definitely conclusive (see Pereira et al., 2018, comment and reply) nor crucial for the tectonic evolution of the region.

4. Incomplete sampling? The Pulo do Lobo, Ribeira de Limas and Santa Iría formations crop out from west to east (from Portugal to Spain) all along the Pulo do Lobo domain, without significant differences along strike. Hence, sampling latitude is irrelevant. We have sampled two transects which include the three formations of the Pulo do Lobo domain (they have been specified as requested; lines 234-239). Thus, we consider unfounded claiming that our sampling was incomplete. Furthermore, some of our samples (PLB-91 and 93) were collected from the same area where the presence of "lawsonite pseudomorphs" was quoted by Rubio Pascual et al. (2013). Concerning the "lousy precision" of our sampling sites (Fig. 1b), we already added UTM coordinates as supplementary information to the first version of the manuscript. Regarding the metamafic rocks, our study was intentionally focused on the metapelites because they better recorded the successive episodes of deformation and, with the new techniques used, the PT conditions of the low-grade metamorphism can also be unveiled with relative accuracy. Future studies focused on the metabasites will be welcome, though keeping in mind that the age of these rocks is early Carboniferous (here again, zircon populations are robust and highly concordant, particularly SHRIMP data), which, in turn, precludes getting information on the early (Devonian) metamorphism of the Pulo do Lobo domain.

5. Sample preparation and interpretation of XRF data. The preparation of oriented aggregates of both whole-rock and <2 $\mu$m fractions strictly followed a well stablished

international procedure to minimize detrital mica content. Obviously, direct discrimination at small grain sizes is a hard task. In our case, both sample fractions yielded similar results, thus suggesting (detrital) mica re-equilibration during M1 metamorphism. Concerning the mechanical rotation of pre-existing minerals, our textural observations and chemical data (Figs. 2 and 3) support the preservation of variably rotated M1 micas during D2, as already discussed in our first manuscript (section 5.1).

6. Lawsonite? The presence of lawsonite in the Pulo do Lobo domain is only based on the external rhomboidal shape of an aggregate of epidote crystals interpreted as a lawsonite pseudomorph (Rubio Pascual et al., 2013). However, lawsonite can be easily mistaken for other minerals, such as clinozoisite. Indeed, the phengites analyzed by these authors "were not particularly indicative of HP recrystallization". Clearly, the lawsonite pseudomorph is an extremely weak evidence for HP metamorphism, which cannot defy our much more complete metamorphic study. Anyway, the claim for a first lawsonite-bearing mineral assemblage was already cited in our first manuscript (section 5.4).

7. Biotite? According to the peak temperatures obtained in our study, biotite could be present in the Pulo do Lobo Fm. Actually, Rubio Pascual et al. (2013) reported two chemical analyses corresponding to biotite. However, we did not identify biotite in any of our samples, possibly because this mineral is restricted to some –and scarce– particular lithologies. This issue has been included in the discussion (lines 567-570).

8. Sedimentary mélange or tectonic mélange? Our study (see also Pérez-Cáceres et al., 2015) suggests that the so-called Peramora mélange is a sedimentary mélange dominated by mafic olistholites, which were later affected by thrusting, thus resulting also in a tectonic mélange (reference to Apalategui et al., 1983 has been deleted here, as requested; line 214). Despite the reviewer's criticism, this twofold characterization of the Peramora outcrop has no contradiction. At other outcrops, the metamafic rocks seem to be intrusive in the Pulo do Lobo Fm. We realize, however, that the description of the Peramora rocks was a bit confusing and have rewritten it (lines 210-211).

9. Deformations recorded by the Santa Iría Fm. First of all, it is important to point out that we agree with the reviewer on two issues: i) three penetrative deformations affected the Pulo do Lobo and Ribeira de Limas Fms of the Pulo do Lobo domain; and ii) Santa Iría Fm unconformably overlies the Pulo do Lobo and Ribeira de Limas Fms. However, the reviewer believes that the Santa Iría Fm is only affected by the third deformation, while we state that it is affected by the last two of them (Fig. 2c). Regarding this issue, the reviewer's argument seems a bit confusing, being apparently based on: i) "it exists a general agreement"(?); and ii) the ages of deformed and undeformed facies of the Gil Márquez pluton (we deal with the Gil Márquez pluton in the following point of our answer). By contrast, our statement that the Santa Iría Fm is affected by two of the three penetrative deformations in the Pulo do Lobo domain is supported by a detailed structural analysis, which includes: cleavage identification, deformational microstructures, local vergences and indicators of stratigraphic polarity all across the Pulo do Lobo domain; the resulting macrostructure is displayed in Fig. 1 (see also Pérez-Cáceres et al., 2015).

10. Timing of deformations and associated metamorphism in the Pulo do Lobo domain. Regarding timing of deformations, the reviewer bases his argument on the ages and deformations recorded by the Gil Márquez pluton. According to him, the lack of deformation in the older mafic facies (354 Ma) implies that the two first penetrative deformations in the Pulo do Lobo domain are older than 354 Ma and therefore they did not affect the Santa Iría Fm. However, the mafic rocks of the Gil Márquez pluton constitute a highly competent body, which is a poor marker for superposed deformations, as demonstrated by the fact that a younger felsic facies of this pluton (345 Ma) is foliated while the older one is isotropic. By contrast, our statements about the Santa Iría Fm are: i) its age is early Carboniferous according to discussion in point 3; and ii) it is affected by two penetrative deformations, as demonstrated by a detailed and comprehensive structural analysis (see point 9). Thus, only the first penetrative deformation of the Pulo do Lobo domain is Devonian (Late Devonian) in age, while the two subsequent deformations are Carboniferous (as also noted previously by Silva et

al., 1990; line 211). Accordingly, the same timing applies to the syn-kinematic meta-morphism associated with the first and second penetrative deformations of the Pulo do Lobo domain.

Regarding the third deformation (D3; upright folding), we already described a spaced and disjunctive crenulation cleavage S3 (lines 189-191, 202), which did not entail phyl-losilicate growth (lines 399-400). This has been stressed in section 5.1 (lines 510-511), and no further implications are attained about D3.

11. The Gil Márquez pluton as evidence of active subduction. The calc-alkaline geo-chemistry of the Gil Márquez rocks is taken by the reviewer as evidence of active subduction at early Carboniferous time. The reviewer views this as a support to the subduction-related accretionary prism interpretation of the Pulo do Lobo domain. We believe, however, that these geochemical features are overrated when used to make inferences of active subduction at early Carboniferous time. Actually, the volume of the Gil Márquez rocks is very scarce to attest a magmatic arc, and, more importantly, the calc-alkaline geochemistry may be obtained from the residual mantle contaminated by the Devonian subduction, which at the time of Gil Márquez intrusion was no longer active. Indeed, there is regional evidence that collision started at Late Devonian time (e.g. Ponce et al., 2012 and references therein).

12. Writing corrections have been introduced in the revised version of the manuscript (lines 103, 136, 143, 234, 562).

To sum up and leaving aside the particular questions addressed above, the main con-cern of the reviewer is about the identification of the Pulo do Lobo lower formations (Pulo do Lobo and Ribeira de Limas Fms) with a subduction-related accretionary prism, an interpretation that we defy in our paper. If not used in a vague way (a usage that we do not endorse), an accretionary prism is a tectonic unit made up of a package of imbrications, having at least one of the two following key features: i) HP meta-morphism indicative of subduction; ii) tectonic slices of mafic rocks and ocean floor

sediments scrapped off from oceanic crust. None of these features is present in the lower formations of the Pulo do Lobo domain, according to the structural, radiometric and metamorphic data reported in our paper. Accordingly, we claim that: i) the structure is dominated by folding, despite local imbrications (see Fig. 1.c.2); ii) the mafic rocks embedded in the Pulo do Lobo Fm are dated at early Carboniferous age, thus being imbricated with Middle-Upper Devonian sediments and not representing slices of oceanic crust; and iii) the metamorphic gradient of these rocks is of low-pressure, as demonstrated by our detailed metamorphic study. Therefore, we maintain our conclusion that the Pulo do Lobo lower formations constitute a tectonic unit located very near the subduccion/collision OMZ/SPZ boundary, but without the most typical features of a subduction-related accretionary prism.

References in this reply: Apalategui, O., Barranco, E., Contreras, F., Delgado, M., and Roldán, F. J.: Hoja 916, Aroche, Mapa Geológico de España a escala 1:50000, Inst. Geológico y Minero de España, Madrid, 1983. Braid, J. A., Murphy, J. B., Quesada, C., and Mortensen, J.: Tectonic escape of a crustal fragment during the closure of the Rheic Ocean: U–Pb detrital zircon data from the Late Palaeozoic Pulo do Lobo and South Portuguese zones, southern Iberia, Journal of the Geological Society, 168(2), 383-392, 2011. Lopes, G., Pereira, Z., Fernandes, P., Wicander, R., Matos, J.X., Rosa, D., and Oliveira, J.T.: The significance of reworked palynomorphs (middle Cambrian to Tournaisian) in the Visean Toca da Moura Complex (South Portugal). Implications for the geodynamic evolution of Ossa Morena Zone, Rev. Palaeobot. Palynol., 200, 1-23, 2014. Pérez-Cáceres, I., Martínez Poyatos, D., Simancas, J.F., and Azor, A.: The elusive nature of the Rheic Ocean in SW Iberia, Tectonics, 34, 2429-2450, 2015. Pérez-Cáceres, I., Martínez Poyatos, D., Simancas, J.F., and Azor, A.: Testing the Avalonian affinity of the South Portuguese Zone and the Neoproterozoic evolution of SW Iberia through detrital zircon populations, Gondwana Research, 42, 177-192, 2017. Pereira, Z., Fernandes, P., Matos, J., Jorge, R., and Oliveira, J.T.: Stratigraphy of the Northern Pulo do Lobo Domain, SW Iberia Variscides: A palynological contribution, Geobios, 51, 491-506, 2018. Pereira, Z., Fernandes, P., Matos, J. X., Jorge, R. C.

and Oliveira, J.T.: Reply to "Comment on Âństratigraphy of the Northern Pulo do Lobo Domain, SW Iberia Variscides: A palynological contributionÂż by Zélia Pereira et al. (2018)–Geobios 51, 491–506". Geobios, 55, 107-110, 2019. Pereira, M.F., Martínez Poyatos, D., Pérez-Cáceres, I., Gama, C., and Azor, A.: Comment on "Stratigraphy of the Northern Pulo do Lobo Domain, SW Iberia Variscides: A palynological contribution" by Pereira, Z. et al. (2018) - Geobios, 51, 491-506. Geobios, in press, 2019. Ponce, C., Simancas, J.F., Azor, A., Martínez Poyatos, D.J., Booth-Rea, G., and Expósito, I.: Metamorphism and kinematics of the early deformation in the Variscan suture of SW Iberia, Journal of Metamorphic Geology, 30(7), 625-638, 2012. Silva, J. B., Oliveira, J.T., and Ribeiro, A.: South Portuguese Zone, estructural outline, in: Pre-Mesozoic Geology of Iberia, edited by: Dallmeyer, R.D., and Martínez García, E., Springer, Berlin, Germany, pp. 348-362, 1990.
* * *

---

## Author Comment (AC2) · 16 Dec 2019

Reply to the interactive comment of referee#2 on the paper entitled "Deciphering the metamorphic evolution of the Pulo do Lobo metasedimentary belt (SW Iberian Variscides)" by Irene Pérez-Cáceres et al. (Manuscript number se-2019-143).

1. General comments: We acknowledge the revision and positive comments of the anonymous referee 2. We also appreciate his/her constructive suggestions, which have contributed to improve the revised version of the manuscript.

2. Specific comments: All of the suggestions and corrections have been attended, as explained below point-by-point (line numbers in brackets correspond to the revised version of the manuscript with tracked changes (uploaded as supplement file to this response): Lines 48-49: the range of celadonite content and average data of b-cell dimension have been included in the abstract (lines 49-50 in the revised manuscript) and in the second paragraph of section 4.2 (lines 426, 431-432). As for the keywords, "X-Ray diffraction" has been substituted by "Illite crystallinity". Line 100 (103): "allows to know" instead of "allows know", as suggested. In lines 163, 167 and 181 (174, 178, 192), "which" has been included after each formation is named, since these sentences are subordinated. Line 228 (244-245): "SEM" has been changed to "scanning electron microscope (SEM)", because this is the first time that is mentioned. Line 276 (292-293): we refer here to the temperature of formation of white-mica, and it has been included according to the suggestion made by the referee. Lines 292-294 (308-310): minerals and synthetic oxides have been rewritten with lowercases in the revised version. Lines 318-321 (lines 337-339): regarding the EPMA analysis at the University of Huelva, information about the standards used and the analysis time has been incorporated in section 3.2. Lines 324-330 (342-348 in the revised version) have been shortened following the referee's suggestion. In addition, "carbonaceous material" has been abbreviated to "CM". Lines 341-344 (360-361): we have included justification of sample rejection for analysis. Line 349 (368): "equipped" instead of "equiped". Line 361 (379): "Acording to SEM analysis" has been replaced by "According to the petrographic study". Line 369-370 (lines 387-388): The last sentence in the introduction to section 4 has been deleted, as suggested. Line 373 (391): "of K-white mica" has been included. Lines 374-376: The referee suggests adding a column in Table 1 with the KI values corresponding to the bulk fraction of the samples. Actually, this column was already included in the first version. Lines 393-394 (411-412 of the revised manuscript): we agree with the referee comment. "very low to" and "presence of C/S and" have been deleted according to the referee's correction. Line 395 (413-414): we have specifically clarified the term "low-pressure gradient" according to Guidotti and Sassi (1986). Also clarified at the end of the abstract ("low pressure/temperature gradient"; line 51). Line (428): "illite" has been replaced by "illitic-mica". Line 412 (431-432): the values of low b-cell parameter and high d001 have been indicated. Lines 419-420 (438-439): an assertion and reference have been incorporated to justify why poor-sudoite chlorites are related to higher temperatures. Line 444 (463): the average temperature range has been revised. Line 445-447 (465-467): the sentence "Nevertheless, the Bourdelle thermometer predicts temperatures up to 380-400°C" has been deleted because it is not significant as the referee explains. Other writing suggestions have also been taken into account. Line 455 (476): Fig. 3 is now cited here. Line 469 (489-492): the sentence has been changed as the referee suggests. Lines 471-473 (492-494): location of Table 2: the last sentence of the first paragraph in section 5.1 has been moved to section 5.3 (600-602) as the referee suggests. Line 481 (502): "our" has been substituted by "these". Line 498 (520-521): according to Abad et al. (2006), the KI value 0.14 $\Delta°2\theta$ is the limit of high epizone conditions. Line 512 (535-536): temperature ranges have been revised, giving the overall range based on the three approaches. Line 513 (537): Fig. 4d is cited now. Line 529 (552): "is difficult" has been substituted by "is really difficult". Lines 531-532 (554-555): "white mica crystallinity and" has been deleted according to the referee's revision. Lines 554-561 (597-604): In these sentences, there are some references on how igneous bodies (especially dykes or plutons) can influence the CM temperatures. The referee asks whether or not dykes really crop out in the studied area. As stated some lines above in the text (and in the geological description), the Pulo do Lobo contains layered mafic intrusions and some granitic plutons that could have enhanced the CM temperature in some samples (eg., sample PLB-93; see also lines 514-519 of the first version). Line 567 (593): potassium feldspar (Fsp) has been incorporated. Figure 1 caption (687): "and collected samples" has been added. Figure 7: the referee asks why the width of the columns in Fig. 7a and Fig. 7b are different; the software used unfortunately does not allow changing it. Table 1 and Figure 1: colour bar: The colour bars ("cold" to "hot" colours as temperature increases) have the only purpose of visual appearance of relative temperatures. To avoid confusion, different coloured bars were used for KI (from green to orange) and

RSCM (from blue to red) parameters. Table 1 caption: "Basel" has been incorporated to "KI values" (732). "Feldespar" has been substituted by "Feldspar" (734). Table 1: the number of decimals are now equal. "KI colour" of sample 82 has been corrected.

References in this reply: Abad, I., Nieto, F., Gutiérrez-Alonso, G., Campo, M. D., López-Munguira, A., and Velilla, N.: Illitic substitution in micas of very low-grade metamorphic clastic rocks. European Journal of Mineralogy, 18(1), 59-69, 2006. Guidotti, C.V., and Sassi, F.P.: Classification and correlation of metamorphic facies series by means of muscovite b data from low-grade metapelites, Neues Jahrbuch fur Mineralogie-Abhandlungen, 153, 363-380, 1986.

Please also note the supplement to this comment:
https://www.solid-earth-discuss.net/se-2019-143/se-2019-143-AC2-supplement.pdf

**Supplement:**

[revised manuscript text omitted]

---

## Editor Decision (ED1)

Minor mistakes on references

- On line 296 the reference is "Andrade et al., 2006; but on line 854 the reference is "De Andrade et al., 2006".
- On line 133 there is a reference "Munhá et al., 1986" which is not in the list.
- "Vidal  et al., 2001" on line  1087: I think is not in the text.

---

## Author Response (AR2)

**Author's response to the second review of the manuscript entitle "Deciphering the metamorphic evolution of the Pulo do Lobo metasedimentary domain (SW Iberian Variscides)" by Pérez Cáceres *et al.* (Manuscript number se-2019-143):**

We acknowledge the positive overview and constructive comments raised by Prof. J.B. Murphy, as well as his formal corrections that contributed to clarify the final manuscript. All the suggestions and corrections have been attended, which will increase the prominence and dissemination of our paper. Below the marked-up revised manuscript version is included.

[revised manuscript text omitted]